# Atmospheric oxygen regulation at low Proterozoic levels by incomplete oxidative weathering of sedimentary organic carbon

Stuart J. Daines[1], Benjamin J. W. Mills[1,2] & Timothy M. Lenton[1]

It is unclear why atmospheric oxygen remained trapped at low levels for more than 1.5 billion years following the Paleoproterozoic Great Oxidation Event. Here, we use models for erosion, weathering and biogeochemical cycling to show that this can be explained by the tectonic recycling of previously accumulated sedimentary organic carbon, combined with the oxygen sensitivity of oxidative weathering. Our results indicate a strong negative feedback regime when atmospheric oxygen concentration is of order $pO_2 \sim 0.1$ PAL (present atmospheric level), but that stability is lost at $pO_2 < 0.01$ PAL. Within these limits, the carbonate carbon isotope ($\delta^{13}C$) record becomes insensitive to changes in organic carbon burial rate, due to counterbalancing changes in the weathering of isotopically light organic carbon. This can explain the lack of secular trend in the Precambrian $\delta^{13}C$ record, and reopens the possibility that increased biological productivity and resultant organic carbon burial drove the Great Oxidation Event.

[1] Earth System Science Group, Department of Geography, College of Life and Environmental Sciences, University of Exeter, Laver Building (Level 7), North Parks Road, Exeter EX4 4QE, UK. [2] School of Earth and Environment, University of Leeds, Leeds LS2 9JT, UK. Correspondence and requests for materials should be addressed to T.M.L. (email: t.m.lenton@exeter.ac.uk).

Atmospheric oxygen (pO$_2$) rose from $< 10^{-5}$ present atmospheric level (PAL) at the Great Oxidation Event, as constrained by the disappearance of mass-independent fractionation of sulfur isotopes[1] ~ 2.45–2.32 Ga. Oxygen levels were sufficient to oxygenate some deep ocean basins[2,3] and deposit gypsum[4,5] during the 'Lomagundi' carbon isotope excursion[2-7] ~ 2.22–2.06 Ga. Subsequently, paleosol oxidation state[8] at ~ 1.85 Ga and ~ 1.1 Ga indicates pO$_2$ > 0.01 PAL, the absence of detrital pyrite and uraninite after ~ 2.1 Ga suggests[9] pO$_2$ > 0.05 PAL, and redox proxies of widespread deep ocean anoxia suggest pO$_2$ < 0.5 PAL (assuming present ocean nutrient levels) until at least the latest Neoproterozoic[10] and probably the mid-Palaeozoic Era[11-13]. Transient Proterozoic pO$_2$ excursions cannot be ruled out by sparse data, and it has recently been suggested from a lack of chromium isotope fractionation in iron-rich sedimentary rocks[14] that pO$_2$ < 0.001 PAL during 1.8–0.8 Ga. This challenges previous inferences[10,15-17] that Proterozoic pO$_2$ was ~ 0.01–0.1 PAL, themselves based on models of paleosol oxidation[8] and of benthic sulfide oxidation[18] that can be questioned. It is in turn challenged by subsequent work inferring pO$_2$ > 0.04 PAL at ~ 1.4 Ga from a lack of vanadium retention in marine sediments[19-21], and work showing chromium isotope fractionation in carbonates ~ 1.1 Ga onwards[22]. Atmospheric oxygen eventually approached modern levels[6,11] during ~ 0.6–0.4 Ga, with the first compelling evidence for pO$_2 \geq 0.6$ PAL from the appearance of Palaeozoic fossil charcoal[23] at ~ 0.4 Ga. Thus, despite uncertainty about the precise levels, it is clear that atmospheric oxygen was somehow trapped at much lower levels than today for > 1.5 Gyr following the Great Oxidation Event.

Identifying the mechanisms responsible for stabilizing oxygen well below present levels poses an outstanding puzzle[24-27]. In particular, the residence time of atmospheric oxygen (~ 8 Myr at pO$_2$ ~ 1 PAL), is short relative to geological timescales. Hence, to explain the long-term stability of pO$_2$ requires negative feedback operating on the oxygen source(s) and/or sink(s). In the Phanerozoic, the major oxygen source is organic carbon burial (derived from oxygenic photosynthesis) and the largest sink is oxidative weathering of uplifted sedimentary organic carbon (kerogen). Models for Phanerozoic oxygen regulation[11,28] tend to focus on strong negative feedbacks on organic carbon burial[29] (that is, that the production rate of oxygen decreases as oxygen rises), and any negative feedback on oxidative weathering is estimated to be relatively weak[30,31] (or non-existent[28,32]). This is because near modern pO$_2$ (~ 1 PAL), uplifted organic carbon is completely oxidized in slowly eroding soil environments[32], meaning oxidative weathering is transport-controlled by the supply of reduced material, rather than kinetically controlled by the pO$_2$ level. However, at high erosion rates, oxidative weathering is incomplete and detrital kerogen is preserved in marine sediments[33-35], meaning that the global oxidative weathering flux is somewhat sensitive to pO$_2$ variations even at the present day[30,31]. Here, we argue that the balance of mechanisms that regulated atmospheric oxygen at much lower Proterozoic levels was quite different, and that incomplete oxidative weathering played a key role. Furthermore, we show that this mechanism makes the carbonate carbon isotope ($\delta^{13}$C) record insensitive to changes in organic carbon burial rate, thus explaining the lack of secular trend in the Precambrian $\delta^{13}$C record.

## Results

**Sedimentary reduced matter cycling.** Our model for the regulation of atmospheric oxygen at low Proterozoic levels hinges on an explicit consideration of the accumulation and tectonic recycling of sedimentary reduced matter over Earth history

(Fig. 1). Importantly, the recycling of sedimentary rocks through erosion, sedimentation and uplift, is quantitatively greater than their conversion by metamorphism, with a ratio of bulk rock fluxes for the modern Earth[36] of ~ 3.5:1 (Fig. 1a). Estimates of the potential sink of oxygen from oxidative weathering (based on the organic carbon, total sulfur and total iron content of sediments) are much larger than the sink from oxidizing volcanic/metamorphic reduced gases (Table 1).

In the modern oxygenated environment (Fig. 1d), much of the sedimentary organic carbon that is uplifted is oxidized[32], with the remainder returning to the sediments as 'detrital' organic carbon[37] (indicated by the thin looped black arrow going from and to the sediments in Fig. 1d). Thus today, organic carbon burial (and corresponding oxygen production) is balanced mostly by oxidative weathering of sedimentary organic carbon (~ $3.75 \times 10^{12}$ mol yr$^{-1}$) with a smaller contribution (~ $1.25 \times 10^{12}$ mol yr$^{-1}$) from oxidation of reduced metamorphic and volcanic gases[11,38] (and an uncertain but potentially comparable contribution from oxidation of thermogenic methane; Table 1, addressed below). Oxidation of sulfides and ferrous iron in sediments and the upper crust are smaller sinks at present, because less than half of the sedimentary iron and sulfur are in reduced form.

Prior to the Great Oxidation Event under pO$_2 < 10^{-5}$ PAL (Fig. 1b), the oxidation of reduced sedimentary organic carbon would have been negligible, with oxygen stabilized instead by the oxygen sensitivity of reactions with reduced gases. Hence, reduced sediments would have been recycled repeatedly until they were deeply buried and metamorphosed. The size of the sedimentary organic carbon reservoir would then have been determined by the balance between input from the burial of newly generated organic carbon, and output via metamorphism. Given the relatively low rate of metamorphic conversion, even a modest rate of 'new' organic carbon burial would have allowed a large sedimentary organic carbon reservoir to accumulate during the Archaean[6,39].

After the Great Oxidation Event (Fig. 1c), the rise of atmospheric oxygen would have enabled the oxidative weathering of uplifted sediments. We assume by this time the majority of organic carbon was produced by oxygenic photosynthesis. The Great Oxidation Event itself was triggered by a secular evolution from net reduced to net oxidized atmospheric inputs[27,40-42]. Therefore, the oxygen source required to trigger it need only have been slightly greater than the supply of reduced gases to the atmosphere and only a fraction of the tectonic supply of reductant in uplifted sediment. In this regime, redox balance requires that oxidative weathering of uplifted sediment is incomplete (limited by global oxygen supply), and the atmospheric oxygen level is determined by the land-surface integrated oxygen-sensitive kinetics of oxidative weathering.

**Oxidative weathering.** Atmospheric oxygen in the aftermath of the Great Oxidation Event should therefore have been stabilized by the oxygen sensitivity of oxidative weathering. The resulting oxygen level and negative feedback strength would have depended on the kinetics of oxidative weathering at low pO$_2$, which are determined by oxygen transport and reaction in soils and regolith, integrated over the continental surface. To quantify this, we used an existing reaction-transport model[32,43] for oxidative weathering of organic carbon and pyrite and ran it repeatedly to span the wide range of erosion rates observed across the continental surface today[44] (see 'Methods' section). Then we obtained a global oxidative weathering flux by weighting these results by the observed fractional areal contributions of different erosion rates across the continental surface[44].

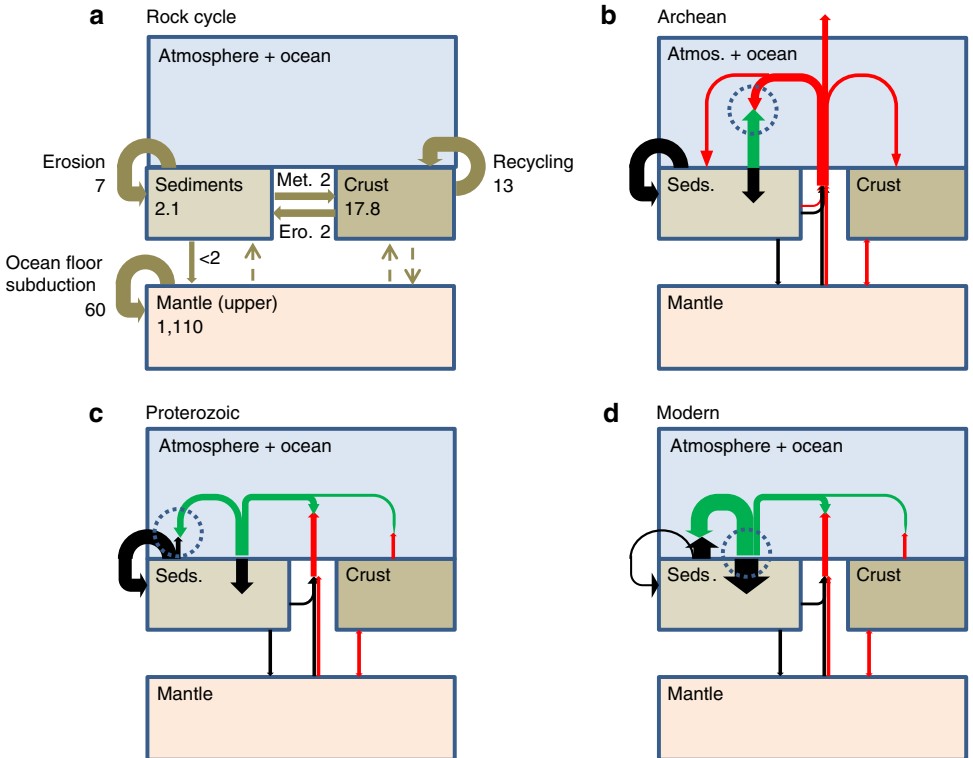

**Figure 1 | Schematic of the rock cycle and the evolution of controls on atmospheric oxygen.** Arrows show fluxes of rock (brown), organic carbon (black), oxygen (green), and other reduced species (red). Dashed circle shows primary negative feedback control on atmospheric oxygen. (**a**) Modern rock cycle[36] fluxes ($10^{18}$ ton Gyr$^{-1}$, 'Met.' = metamorphism, 'Ero.' = erosion) and masses ($10^{18}$ ton). (**b**) Archean: organic carbon burial is balanced by metamorphism, with negligible oxidative weathering. Atmospheric oxygen is a minor component with concentration determined by the oxygen sensitivity of reactions with reduced atmospheric gases. (**c**) Proterozoic: sedimentary organic carbon is partly oxidized but mainly recycled. Atmospheric oxygen is controlled by the oxygen sensitivity of oxidative weathering. (**d**) Modern: sedimentary organic carbon is oxidized with little recycling. Atmospheric oxygen is controlled by feedbacks on carbon burial.

The results (Fig. 2) show that organic carbon (kerogen) is completely oxidized at $pO_2 = 1$ PAL for low and intermediate erosion rates, but is incompletely oxidized at the highest erosion rates today; 50 cm kyr$^{-1}$ (dominated by large islands of the Western Pacific) and 25 cm kyr$^{-1}$ (dominated by the Himalayas) (Fig. 2a, see 'Methods' section). This is consistent with observations of detrital organic carbon from these regions reaching marine sediments[34,35]. It confirms that there is a negative feedback dependence of oxygen removal on $pO_2$ near $\sim 1$ PAL, but it is fairly weak.

As $pO_2$ declines, the oxidative weathering flux declines and becomes more sensitive to $pO_2$ variations, because oxidative weathering becomes kinetically-limited in a wider range of regions with progressively lower erosion rates (Fig. 2a,b). Initially, this amounts to a strengthening of the negative feedback on $pO_2$. At $pO_2 \sim 0.1$ PAL, there is still a significant oxidative weathering flux which is a strong function of oxygen concentration (Fig. 2a,b), giving the potential to provide strong negative feedback on $pO_2$. However, as $pO_2$ declines further, below $pO_2 \sim 0.03$ PAL, although oxidative weathering remains sensitive to $pO_2$ the potential negative feedback becomes weaker as this sink becomes relatively small (Fig. 2b). Overall a dependency of oxidative weathering on $pO_2^{0.5}$ (Fig. 2a, dashed line), consistent with coal oxidation kinetics[45] and as assumed in some previous biogeochemical models[11,25], provides a reasonable fit to the results.

We performed a sensitivity analysis increasing shale oxygen diffusivity (controlled by porosity) and removing the contributions of high-uplift regions to global sediment discharge rates (see 'Methods' section). This can weaken the sensitivity of the global oxidative weathering flux to oxygen variation near $pO_2 = 1$ PAL and shifts the region of strong negative feedback to somewhat lower $pO_2$.

**Oxygen regulation.** The relative strength of the oxidative weathering feedback is determined by its size relative to other sinks (Table 1), especially atmospheric sinks, which would have been insensitive to $pO_2$ at levels after the Great Oxidation Event[40,42] (Fig. 2b). Including the additional sink from metamorphic/volcanic reduced gases demonstrates the dependence of $pO_2$ on organic carbon burial rate (Fig. 3). Here, we first consider parameterizations for the modern Earth and a minimum estimate for atmospheric reductant inputs. Assuming negligible burial of terrestrial organic matter in the Proterozoic, which comprises $\sim 50\%$ of total burial today[11,13,46], and assuming modern ocean nutrient levels, marine organic carbon burial could have been comparable to the modern level of $\sim 2.5 \times 10^{12}$ mol C yr$^{-1}$. Combining this with minimum estimates for volcanic and metamorphic inputs, the model predicts $pO_2 \sim 0.1$ PAL (Fig. 3a) with oxygen stabilized by the negative feedback from oxidative weathering. Varying the distribution of continental erosion rates between plausible limits then causes $pO_2$ to vary by a factor of $\sim 2$, with low global erosion and/or increased shale porosity, and therefore more complete oxidation, producing lower $pO_2$ (Fig. 3a). Varying organic carbon burial can cause larger variations in $pO_2$ within the same stable state. However, if organic carbon burial

**Table 1 | Potential oxygen sinks based on modern-day fluxes.**

| | Content*(wt%) | Flux ($10^{12}$ mol yr$^{-1}$) | Potential sink ($10^{12}$ mol O$_2$ eq yr$^{-1}$) | Precambrian assumptions |
|---|---|---|---|---|
| *Erosion*[†] | | | | |
| C organic, sediments | 0.4–0.6 | 2.33–7.78 | 2.33–7.78 | |
| S total, sediments | 0.381–0.472 | 0.83–2.30 | 1.66–4.60 | All S as $S_2^{2-}$ |
| Fe total, sediments | 3.58–4.49 | 4.48–12.51 | 1.12–3.13 | All Fe as $Fe^{2+}$ |
| Fe total, upper crust | 3.09–3.5 | 1.10–2.78 | 0.28–0.69 | All Fe as $Fe^{2+}$ |
| Total | | | 5.39–16.20 | |
| | | | | |
| *Metamorphism*[‡] | | | | |
| C organic | | 0.67–2.22 | 0.67–2.22 | |
| S total | | 0.24–0.65 | 0.48–1.31 | |
| Total[§] | | | 1.15–3.53 | |
| | | | | |
| *Volcanic reduced gases* | | | | |
| SO$_2$ (refs 42,52) | | 1.0–1.8 | 0.5–0.9 | |
| H$_2$ (refs 42,52) | | 1.0–2.4 | 0.5–1.2 | |
| H$_2$S (ref. 42) | | 0.03 | 0.06 | |
| Total[§] | | | 1.06–2.16 | |
| | | | | |
| *Thermogenic methane* | | | | |
| CH$_4$ (refs 42,65) | | 1.55–3.38[‖] | 3.1–6.76[‖] | See main text |
| | | | | |
| *Seafloor hydrothermal alteration* | | | | |
| Basalt oxidation[48] | | | 1.0 | Small[¶] |
| Serpentinisation[48] | | | 0.2 | |
| Total | | | 1.2 | |

*Range of composition of sediments and upper crust from Wedepohl[66] and Li[67].
[†]Range of total erosion rates 9–20 × $10^{15}$ g yr$^{-1}$ spans estimates for pre-anthropogenic[36] to modern anthropogenic-influenced[68] values, with 7/9 from sediments and 2/9 from upper crust[36].
[‡]Upper estimate assuming transfer of all reducing power from organic C and total S to gaseous form as sediments are converted to upper crust at a rock flux equivalent to erosion of upper crust (that is, neglecting graphite carbon and sulfides in metamorphic rocks).
[§]The reasonable agreement of these totals is consistent with the volcanic reduced gases coming predominantly from high-temperature metamorphism of sediments (rather than the mantle) and therefore these are two estimates of the same oxygen sink.
[‖]These estimates[69] are debated[65] partly because of inconsistency with the global ethane budget[70].
[¶]Seafloor oxygen consumption is largely due to basalt oxidation by sulfate, hence is expected to have been much smaller in a low-sulfate Precambrian environment[48].

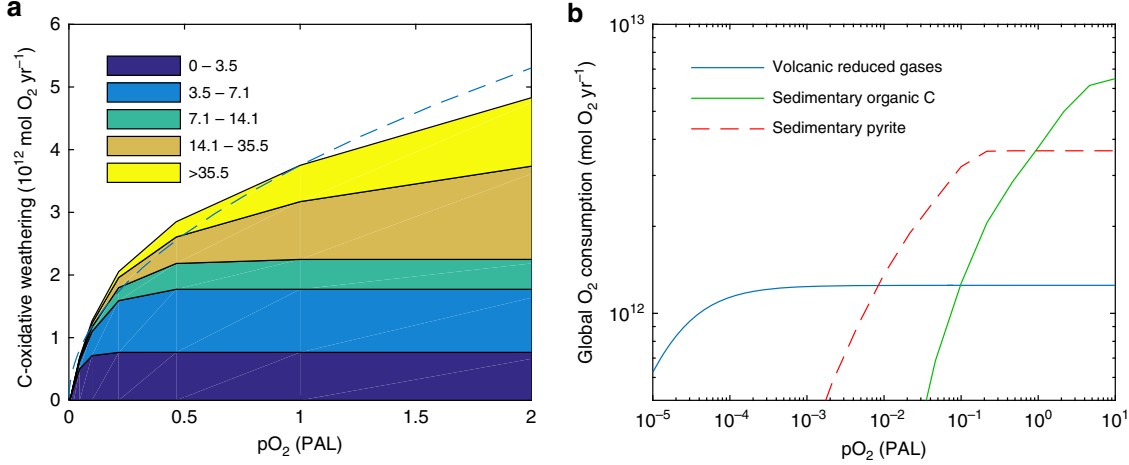

**Figure 2 | Sensitivity of atmospheric oxygen sinks to oxygen concentration.** (**a**) Oxidative weathering rate for carbon, from the shale reaction-transport model of Bolton *et al.*[32] integrated over a distribution of land-surface erosion rates[44], showing contributions to carbon oxidation from bins for erosion rate (cm kyr$^{-1}$). Dashed line shows pO$_2^{0.5}$ dependency[45] assumed in some biogeochemical models[11,25]. (**b**) Global integrated oxygen consumption from volcanic reduced gases (blue), sedimentary organic carbon oxidation (green) and sedimentary pyrite oxidation (dashed red). The organic carbon and pyrite results are from the modelling herein, whereas the volcanic reduced gas consumption is a schematic curve to show pO$_2$ sensitivity at pre-Great Oxidation Event levels ($\leq 10^{-5}$ PAL) but not at Proterozoic levels[40,42] (hence this process cannot help explain Proterozoic atmospheric oxygen regulation).

drops below ∼25% of present (∼$1.25 \times 10^{12}$ mol C yr$^{-1}$), corresponding to pO$_2$ < 0.01 PAL, this state loses stability and a 'Great Deoxygenation' is predicted, because the oxygen source to the atmosphere is insufficient to counterbalance the volcanic input of reduced matter. Conversely, an increase in organic carbon burial from lower values to > 25% of present is sufficient

to trigger the Great Oxidation Event in the model, as the net biological source of oxygen exceeds the volcanic input of reduced matter.

This means that to increase atmospheric oxygen toward modern levels (∼1 PAL) requires a factor of ∼4 larger increase in organic carbon burial flux than needed for the Great Oxidation

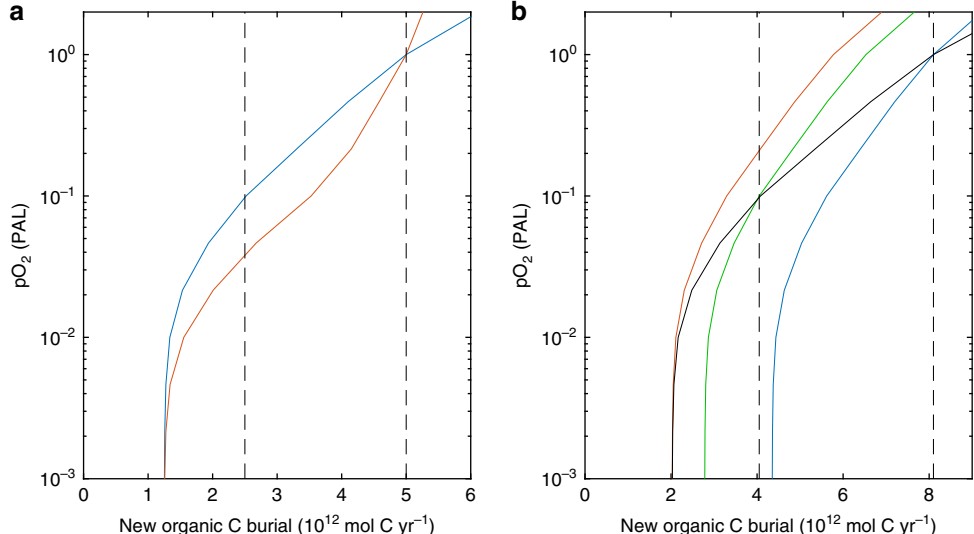

**Figure 3 | Dependence of atmospheric pO$_2$ on organic carbon burial rate.** (**a**) Atmospheric pO$_2$ level at steady-state for oxidative weathering model and minimum estimate $1.25 \times 10^{12}$ mol O$_2$ eq yr$^{-1}$ reduced volatile flux (Table 1). Solid lines show sensitivity to land-surface uplift distribution and shale porosity (see Methods and Fig. 7) with default parameters (blue), and with very low uplift and increased shale porosity ($n^{\star} = 1.5$) (red)—both normalized to give the same pO$_2$ at modern organic carbon burial rates. Vertical dashed lines show modern total and estimated marine (50% of total) organic carbon burial rate. (**b**) Default parameters as in **a**, but including a larger reduced atmospheric flux from a methane metamorphic pathway ($1.55 \times 10^{12}$ mol CH$_4$ yr$^{-1}$; Table 1) and a corresponding increase in organic carbon burial (to $8.1 \times 10^{12}$ mol C yr$^{-1}$), with the methane metamorphic pathway assumed to adjust to 38% of the long-term average carbon burial rate (black line). The other lines show the response with a constant metamorphic methane flux, fixed at 38% of initial burial rates of $8.1 \times 10^{12}$ mol C yr$^{-1}$ (modern; blue), $4 \times 10^{12}$ mol C yr$^{-1}$ (Proterozoic; green) and $2 \times 10^{12}$ mol C yr$^{-1}$ (Great Oxidation Event transition; red). These cases represent the short-term response to perturbations because we presume that on long timescales the metamorphic flux of methane must be proportional to the organic carbon flux previously deposited.

Event (Fig. 3a). This second transition is not as abrupt, but still represents a fundamental change in oxygen regulation regime, in which reductant burial exceeds the reductant supply via sediment recycling and the predominant negative feedback control shifts from the oxygen sink to the oxygen source.

A major uncertainty in atmospheric reductant input is the contribution of thermogenic methane from organic carbon metamorphism (Table 1). If this is assumed to be controlled by overall sedimentary organic carbon content independent of organic carbon burial rate, then this provides a large additional oxygen-independent sink (Fig. 3b, blue line), reducing the relative strength of the oxidative-weathering feedback. However, thermogenic methane is more plausibly linked to a low temperature metamorphic pathway primarily associated with relatively recently buried organic carbon (age $\lesssim 100$ Myr, but longer than the oxygen residence time of $\sim 10$ Myr). In this case it scales with the organic carbon burial rate and hence has a greatly reduced influence in the Precambrian (Fig. 3b, black line), although it still reduces the stability domain of Proterozoic pO$_2$ with respect to short-timescale perturbations (green line).

Secular changes in tectonic (and solar) forcing will modify this picture[40]. As an illustrative case, an 'episodic continental growth' model[47] assumes 80% of present continental area at 2.5 Ga, and predicts global heat flux $Q \sim 1.5$ of present, and ocean crust formation rate and high-temperature hydrothermal heat loss scaling with $Q^2 \sim 2.25$ of present. Metamorphic fluxes plausibly scale proportional to global heat flux ($Q$) and continental area, hence would be $\sim 1.2$ of present (increasing both chemical weathering of phosphorus hence oxygen source, and reductant input hence oxygen sink). Mantle inputs scale as $Q^2$ increasing both CO$_2$ input and the seafloor hydrothermal oxygen sink (primarily serpentinisation in the low-sulfate Precambrian[48]). The overall sign of the combined effect on oxygen level will therefore be model-dependent.

**Effects on the carbon isotope record.** Our proposed mechanism for Proterozoic oxygen regulation changes the interpretation of the Precambrian carbonate carbon isotope ($\delta^{13}$C$_{carb}$) record. Conventionally the constancy of $\delta^{13}$C$_{carb}$ is taken to imply a constant 'f-ratio' of 'new' organic to inorganic carbon burial[49–51]. However, in an oxidative-weathering-limited regime, persistent changes in organic carbon burial result in large counterbalancing changes in oxidative weathering of organic carbon (via changes in pO$_2$). The detail of the transient adjustment process and return to steady-state for an arbitrary decrease (and later increase) in organic carbon burial is illustrated in Fig. 4. Initially there is a drop in $\delta^{13}$C$_{carb}$ as oxidative weathering exceeds organic carbon burial. However, after $\sim 10$ Myr, atmospheric oxygen and oxidative weathering decrease to a new steady state, leaving $\delta^{13}$C$_{carb}$ unchanged from its initial value, with both the net input and output fluxes of $\delta^{13}$C to/from the ocean isotopically heavier than they were (because isotopically-light organic carbon input and output fluxes have declined relative to isotopically heavier inorganic carbon fluxes). Similarly, an increase in organic carbon burial results in a transient increase in $\delta^{13}$C$_{carb}$ and return to its initial value.

The long-term steady-state isotopic composition of the ocean and $\delta^{13}$C$_{carb}$ is therefore independent of the burial rate of 'new' organic carbon (Fig. 5a), as oxygen level adjusts such that net input and output fluxes of $\delta^{13}$C to/from the ocean are equal[39]. Hence, long-term changes in 'new' organic carbon burial during the Proterozoic are not expected to show up in the carbon isotope record. Sedimentary recycling of organic carbon during the Archaean and Proterozoic can thus help reconcile increases in oxygen, and presumed associated increases in biological productivity, with the lack of a secular trend in $\delta^{13}$C$_{carb}$ until the Phanerozoic[17] (where there is a shift from 0 to 2‰ associated with the rise of land plants[13]). As long as the erosion rate is unchanged,

changes in new organic carbon burial change the relative proportion of detrital and new organic carbon being buried (Fig. 5b).

## Discussion

As our focus is on oxygen regulation after the Great Oxidation Event (rather than long-term controls on organic carbon

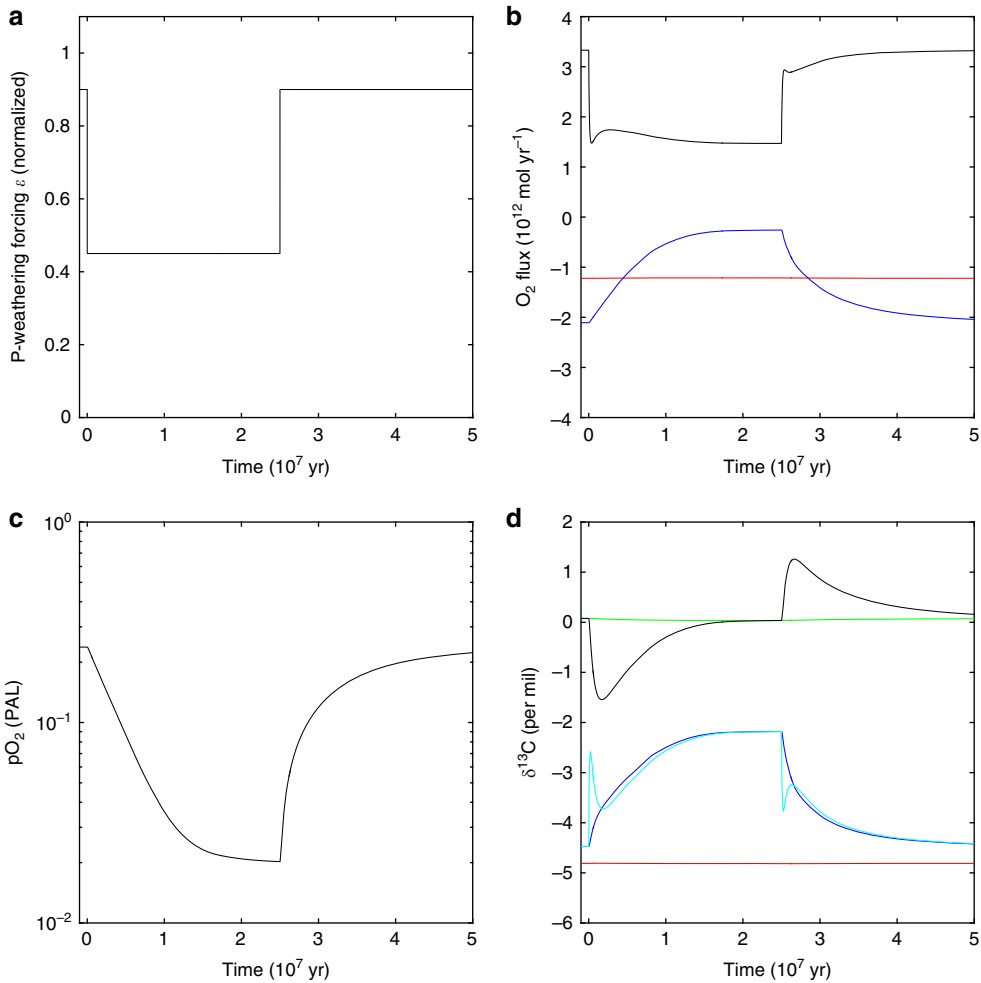

**Figure 4 | Transient response to changes in organic carbon burial flux.** Results from the Precambrian COPSE model with perturbation applied to the model steady-state at 1 Ga (see 'Methods' section). (**a**) Phosphorus weathering perturbation via parameter $\varepsilon$. (**b**) Atmosphere/ocean oxygen fluxes: source from marine organic carbon burial (black), sinks from atmospheric reactions with reduced volcanic/metamorphic gases (red), and oxidative weathering (blue). (**c**) Atmospheric oxygen $pO_2$ showing decrease and approach to new steady-state at 25 My, where oxidative weathering and volcanic sinks again balance production by carbon burial, followed by return to initial level. (**d**) Carbon isotope responses of marine carbonate burial $\delta^{13}C_{carb}$ (black), input (blue) and output (cyan) to ocean/atmospheric system, with mean of sedimentary carbonate carbon (green) and degassing (red). Oxidative weathering initially exceeds organic carbon burial, resulting in a negative $\delta^{13}C$ transient. After ~10 Myr, atmospheric oxygen and oxidative weathering decrease, leaving $\delta^{13}C$ unchanged. Similar behaviour occurs when organic carbon burial is increased again after 25 Myr.

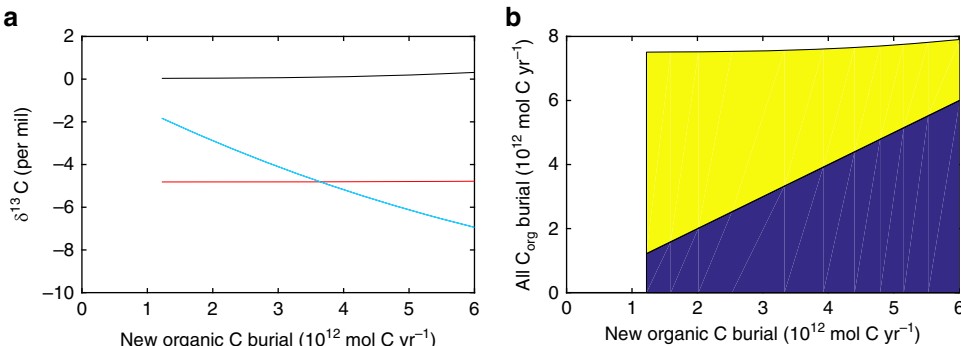

**Figure 5 | Steady-state response of carbon isotopes to changes in organic carbon burial.** Results from the Precambrian COPSE model (see 'Methods' section). (**a**) $\delta^{13}C$ for degassing input (red), carbonate carbon burial (black, $\delta^{13}C_{carb}$) and total inputs/outputs to the surface DIC pool (cyan), which balance one another at steady state. (**b**) New (blue) and detrital (yellow) contributions to total organic carbon ($C_{org}$) burial.

accumulation) we have assumed that the Proterozoic rock cycle was in overall redox balance[51] (as in Phanerozoic carbon cycle models including GEOCARB[38] and COPSE[11]). We have also assumed that atmosphere-ocean reduced input flux is dominated by that from metamorphic sedimentary volatiles, as for the modern Earth. Higher inputs of reductant relative to today—either due to higher mantle fluxes[52], ongoing crustal oxidation during metamorphism hence more reduced metamorphic fluxes[40] or greater submarine volcanism resulting in more reduced volatiles[53]—would affect the stability of an oxygen-sink-controlled feedback regime. However, we suggest these increased reductant inputs had largely declined by the Great Oxidation Event or at latest the end of the Lomagundi excursion. For example, assuming crust oxidation (resulting in a present $Fe^{3+}$ excess[54] of $\approx 2 \times 10^{21}$ mol $O_2$ eq) is linked to continental growth, an 'episodic continental growth' model[55] with rapid growth from 10 to 80% of present area 3.2–2.5 Ga would imply a net oxidation rate of $\approx 2 \times 10^{12}$ mol $O_2$ eq $yr^{-1}$ (balanced by a combination of hydrogen escape and organic carbon accumulation). This would drop after 2.5 Ga to a net oxidation rate of $\approx 1.6 \times 10^{11}$ mol $O_2$ eq $yr^{-1}$, which is small relative to sediment and metamorphic redox transformations (Table 1). However, the iron oxidation state of shales only changes after the Great Oxidation Event, suggesting significant ongoing oxidization of crustal iron[6]. We have assumed that any increase in oxygen source triggering the Great Oxidation Event was modest, but if the oxygen source increased a lot, for example, in the Lomagundi event, then the oxidative weathering sink might have been temporarily overwhelmed leading to high oxygen levels[2–7]. Either way by $\sim 1.85$ Ga oxygen levels were low again.

Our proposed Proterozoic oxygen regulation mechanism requires that marine organic carbon burial remained at least $\sim 25\%$ of today's total organic carbon burial flux after the Great Oxidation Event. Marine-derived organic carbon is estimated to comprise half[11,13] to two-thirds[56] of today's total organic carbon burial flux (the remainder being derived from land plants). Hence, marine organic carbon burial must have remained at least $\sim 40–50\%$ of its present value since the Great Oxidation Event. This in turn requires at least $\sim 40–50\%$ of modern ocean nutrient concentrations, or if they were lower, a more efficient biological carbon pump and/or more efficient sedimentary preservation of organic carbon. Several studies have suggested that nutrient levels were low[25] or the biological pump was less efficient[57] in the Proterozoic. However, if total organic carbon burial was $<25\%$ of present, corresponding to $pO_2 \sim 0.01$ PAL, our model predicts a reversal of the Great Oxidation Event, because although the oxidative weathering of ancient organic carbon would have become negligible, the oxygen source to the atmosphere would have been insufficient to counterbalance the volcanic/metamorphic input of reduced matter. Atmospheric oxygen then drops until the oxidation of these gases becomes kinetically-limited (that is, oxygen concentration limited) at $pO_2 < 10^{-5}$ PAL (Fig. 2b). However, the lack of MIF-S[1] after 2.32 Ga constrains $pO_2 > 10^{-5}$ PAL. Therefore, if recent inferences[14] of Proterozoic $pO_2 < 0.001$ PAL are correct, they require an as yet unidentified oxygen regulating mechanism, which operates somewhere in the range $10^{-5} < pO_2 < 10^{-2}$ PAL.

The sulfur cycle could not have provided an equivalent negative feedback on $pO_2$ at lower levels, even though oxidation and subsequent reduction of sulfur were more important fluxes in the Proterozoic surface redox budget than they are today[15]. This is because the Precambrian sulfur cycle was dominated by a reduced sedimentary reservoir (pyrite), which after surface oxidation in weathering was soon reduced again and buried, whereas the carbon cycle consists of a primarily oxidized reservoir that is available for biotic burial of the reduced form.

This burial of reduced carbon is nutrient (rather than carbon) limited whereas the burial of reduced sulfur only requires a redox gradient generated by the much larger carbon export production flux.

The oxidative weathering model predicts that beneath the upper soil layers, the weathering environment was essentially anoxic and reducing for Proterozoic oxygen levels $pO_2 < 0.1$ PAL (for example, Fig. 6). This has important implications for interpreting redox-sensitive proxies, namely that they are controlled by fluvial transport times and corresponding oxygen exposure[9], not by time spent in the weathering environment (as has erroneously been assumed in previous simpler models[14,58]). For pyrite this means oxidation in soils becomes kinetically-limited at $pO_2 < 0.1$ PAL (Fig. 2b), starting in the most rapidly eroding terrains, and implying a supply of detrital pyrite to fluvial systems. Existing work[9] estimates that for rivers with short transport distances, often found in the most rapidly eroding terrains (for example, on volcanic arc islands), some of this pyrite would survive oxidation at $pO_2 < 0.05$ PAL. Indeed some detrital pyrite survives oxidation today in the most rapidly eroding settings[59]. Therefore recent inferences[14] of $pO_2 < 0.001$ PAL appear inconsistent (by orders of magnitude) with the absence of detrital pyrite in Proterozoic sediments. Again, the one-box model behind these very low $pO_2$ inferences[14] can be questioned because it assumes that manganese can be oxidized in the weathering environment imparting Cr-isotope fractionation above $pO_2 \sim 0.001$ PAL, when instead the soil is predicted to be anoxic and reducing up to $pO_2 \sim 0.1$ PAL (Fig. 6).

Taking into account the incomplete oxidation of organic carbon under low Proterozoic $O_2$, it appears that the lack of a long-term secular trend in $\delta^{13}C_{carb}$ until the Phanerozoic[17] cannot be taken to imply a constant 'f-ratio' of new organic to inorganic carbon burial (as is standard practice[49–51]). Instead there could have been large Precambrian changes in new organic carbon burial that may not show up in long-term $\delta^{13}C_{carb}$, reopening the possibility that, for example, the Great Oxidation Event was driven by an increase in new organic carbon burial. The conventional interpretation of the long-term carbon isotope record (in terms of changing proportions of new organic to inorganic carbon burial) is only appropriate in a high oxygen world where oxidative weathering of organic carbon is fairly

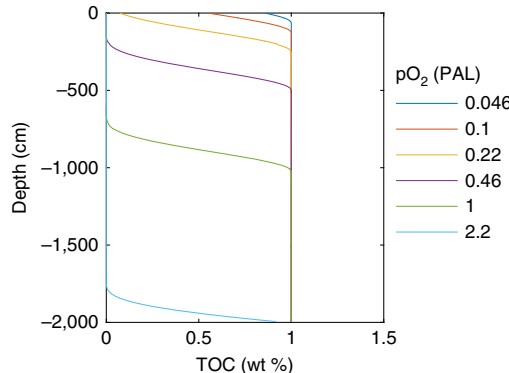

**Figure 6 | Dependence of the organic carbon oxidative weathering horizon depth on $pO_2$.** Results of the shale model assuming default model parameters (see 'Methods' section, Table 2) with initial TOC content of 1 wt%, pyrite S content 0.8 wt% (pyrite not shown), erosion rate of 5 cm $kyr^{-1}$, and values of atmospheric $pO_2$ (PAL) as shown in the legend. The lines show the TOC remaining as a function of depth for different prescribed $pO_2$ (PAL) levels. In each case a distinct weathering horizon forms (above which TOC has been oxidized and below which it has not), with the weathering horizon at greater depth for higher $pO_2$.

complete, and therefore the $\delta^{13}C$ value of carbon inputs remains fairly stable. Even today there is some compensation of changes in new organic carbon burial and pO$_2$ by corresponding changes in the input of ancient organic carbon with the same isotopic composition.

There are of course long-term ($>1$ Myr) $\delta^{13}C_{carb}$ excursions in the Precambrian, notably the Lomagundi event[3,5–7] ($\sim 2.22$– 2.06 Ga). This can be explained by redox exchanges with sedimentary sulfur and iron reservoirs, shifting electrons from sedimentary sulfide and siderite to reduced organic carbon (and back again) on the sedimentary recycling timescale[6,60]. Such $\delta^{13}C_{carb}$ excursions can also arise from changes in the inorganic carbon cycle or in $\delta^{13}C$ fractionation. The carbon isotope decoupling observed in the Neoproterozoic $\delta^{13}C$ record[61] (where $\delta^{13}C_{carb}$ varies but $\delta^{13}C_{org}$ is near constant) can, as previously proposed[49,61,62], be explained by the recycling of detrital organic carbon. Furthermore, the spatially-variable mixing of detrital and marine sources implies that the $\delta^{13}C_{org}$ signature will be location-dependent.

The mechanism we describe provides a potential explanation for the overall stepwise evolution of atmospheric oxygen on Earth. Recycling of sedimentary organic carbon and its kinetically limited oxidation could have sustained low Proterozoic atmospheric oxygen levels—albeit not as low as has recently been inferred[14]. The rise of oxygen to present levels simply required an increase in organic carbon burial rate, which may have begun in the Neoproterozoic Era[10], but in our model was not completed until the mid-Palaeozoic Era with the colonization of the land by plants and fungi, liberating nutrient phosphorus from rocks and producing high C/P material for burial[11,13].

## Methods

**Land-surface oxidative weathering model.** Oxidative weathering of organic carbon and pyrite was simulated using an existing 1D reaction-transport model[32,43] (Table 2), integrated over the observed distribution of uplift rates estimated from river sediment budgets[44].

The 1D shale model represents the steady-state balance between downwards oxygen diffusion in a porous shale and reaction with organic carbon and (optionally) pyrite. It thus quantifies the local transition between a supply-limited regime (where a diffusive oxygen supply exceeds reduced matter supply, a reaction

front forms in the shale column, and all supplied reduced matter is consumed), and a kinetic-limited regime where reduced material is incompletely oxidized with the remainder eroded. The model has been calibrated against laboratory measurements constraining local kerogen[45] and pyrite[32,63] oxidation kinetics, and a field study constraining oxygen transport through porous shale[43]. The shale model was run repeatedly to span the present-day global distribution of erosion rates across the continental surface[44] and the results weighted by the observed fractional areal contributions of these different erosion rates to obtain a global oxidative weathering flux. The estimated total oxidative weathering flux was then scaled to the present day value at pO$_2 = 1$ PAL.

Model parameters (summarized in Table 2) are taken from the default parameter set of Bolton et al.[32] (their Table 1 and Figs 3–5), which generate a reaction front at depth $\sim 10$ m for oxidative weathering of organic carbon at global-mean values for TOC (1 wt%) and erosion rate (5 cm kyr$^{-1}$) (Fig. 6). For given shale properties (organic matter content, porosity, temperature) the reaction front depth is proportional to atmospheric oxygen level and inversely proportional to supply rate (= erosion rate). If the reaction front reaches the surface, organic matter oxidation is incomplete and detrital organic matter is eroded.

The dependency of the land-surface integrated oxidation rate on pO$_2$ is therefore sensitive to oxygen transport, controlled by shale porosity ($n^\star$), and reduced matter supply, controlled by organic carbon and pyrite content and the distribution of uplift rates (Fig. 7). As demonstrated by Bolton et al.[32], at pO$_2 \sim 1$ PAL organic carbon is completely oxidized for uplift rates $<10$ cm kyr$^{-1}$ and even conservative (low porosity; $n^\star = 2$) shale parameters. However, integration over a distribution of uplift rates results in a land-surface average with incomplete oxidation in high-uplift regions, which is very different to that of a homogeneous surface with mean uplift rate ($\sim 5$ cm kyr$^{-1}$).

In a sensitivity analysis (Fig. 7) we examined how the oxygen dependence of global organic carbon oxidation varied with removing the contribution to global sediment discharge rates[44] of regions with the highest uplift (oceanic islands), or those plus the next highest uplift regions (SE Asia and the Himalayas), and with increasing shale oxygen diffusivity ($n^\star = 1.5$). These changes were designed to maximize O$_2$ uptake and thus establish a lower limit at which this mechanism could regulate atmospheric O$_2$.

**Precambrian COPSE model.** We parameterize the COPSE model[11] for the Precambrian[64] (sufficient to represent the oxygen controls and carbon isotope response discussed in the main text) with the following minimum set of changes from the original model[11] to represent a high-CO$_2$ system with no land biota and reduced solar forcing:

The rates of organic carbon burial and oxidation were set as discussed in the main text (organic carbon burial $5 \times 10^{12}$ mol C yr$^{-1}$, oxidative weathering $3.75 \times 10^{12}$ mol C yr$^{-1}$, metamorphic/volcanic reductant input $1.25 \times 10^{12}$ mol C yr$^{-1}$), to be consistent with estimates from sediment erosion rates, with carbon to sulfur ratios, and with arguments for the relative sizes of the organic carbon metamorphic and weathering sinks. Rates are comparable to those in the GEOCARB series of models[38] and are reduced by approximately a factor-of-two relative to the original COPSE model[11].

**Table 2 | Parameters for the land-surface oxidative weathering model (from Bolton et al.[32] 'default model run' except where noted).**

| Quantity | Value | Units | Notes |
|---|---|---|---|
| Organic matter content | 1 | TOC % by mass | Lower boundary condition |
| $d$ (grain thickness) | 10 | μm | OM grain thickness |
| $\alpha$ | 5 | Dimensionless | OM width/thickness |
| $R_{max}$ | $1.015 \times 10^{-4}$ | mol C m$^{-2}$ yr$^{-1}$ | OM Michaelis–Menten rate parameter, 24 °C |
| $K_m$ | $1.787 \times 10^{-6}$ | mol O$_2$ cm$^{-3}$ | OM Michaelis–Menten parameter |
| $Ea_{OM}$ | 42 | kJ mol$^{-1}$ | Activation energy for organic matter oxidation T dependence |
| Pyrite S content | 0.8* | S wt% | Lower boundary condition |
| $d$ (grain thickness) | 20 | μm | Pyrite grain thickness |
| $\alpha$ | 1 | Dimensionless | Pyrite width/thickness |
| $R_{pyr}$ | $2.013 \times 10^{-12}$ | mol Pyr m$^{-2}$ s$^{-1}$ | Pyrite oxidation rate, at [O$_2$] = 1μM l$^{-1}$, 25 °C |
| | 0.6696 | Dimensionless | Pyrite power-law oxygen dependence |
| $Ea_{pyr}$ | 50 | kJ mol$^{-1}$ | Activation energy for pyrite oxidation T dependence[63] |
| Temperature | 15† | °C | |
| $\phi$ (Total porosity) | 0.05 | Dimensionless | Initial pore volume/bulk volume |
| $s_{air}$ | 0.9 | Fraction | Air volume/pore volume |
| $n^\star$ | 2 | Dimensionless | Tortuosity parameter |
| Domain depth | 20 | m | |
| Number of depth bins‡ | 380 | Dimensionless | 1 cm bins for top 2 m, 10 cm bins for 2–20 m |

*Pyrite S content chosen to give organic carbon:S ratio equal to the present-day sedimentary organic carbon: total S ratio ($1.25 \times 10^{21}$ mol C: $0.38 \times 10^{21}$ mol S from Bergman et al.[11]), that is, present-day organic C and total S with all S assumed as pyrite for the Precambrian.
†Present-day global-mean temperature, cf 17 °C used by Bolton et al.[32].
‡Bin size decreased to 1 cm for top 2 m to resolve oxidation front at low pO$_2$.

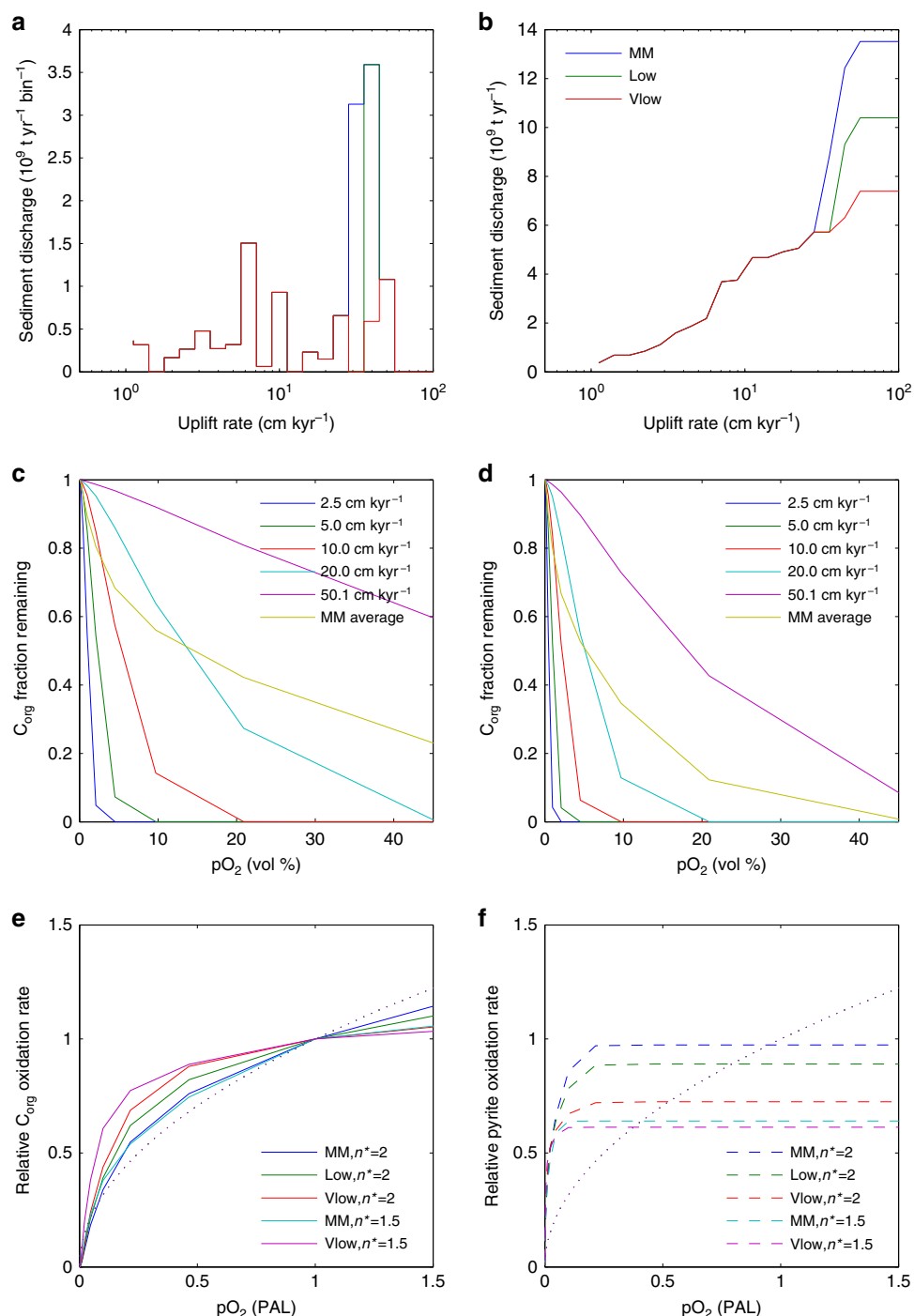

**Figure 7 | Sensitivity analysis of oxidative weathering model. (a)** Distribution of global sediment discharge rates from Milliman and Meade[44] ('MM') versus uplift rate (logarithmic bin width). Contributions from high-uplift regions SE Asia and Himalayas (blue) and oceanic islands (green) are identified. **(b)** Data as **a** showing cumulative global sediment discharge rate versus uplift rate and highlighting the large contribution to global total from high-uplift regions: global total ('MM'; blue), omitting oceanic islands ('low'; green), omitting SE Asia and Himalayas and oceanic islands ('vlow'; red). **(c)** Fraction of organic carbon ($C_{org}$) remaining at surface as a function of atmospheric oxygen level for default shale model parameters ($n^* = 2$), for a range of uplift rates (cm kyr$^{-1}$), and for the land-surface average for modern distribution of uplift rates ('MM average'). **(d)** As **c**, for increased shale oxygen diffusivity ($n^* = 1.5$). **(e)** Sensitivity of oxygen dependence of global organic carbon oxidation rate to shale diffusivity ($n^*$, two values as for **c,d**) and uplift rate distribution (as in **a,b**). Key for uplift rate distribution: 'MM', modern global total; 'low', omits oceanic islands; 'vlow', omits oceanic islands and SE Asia and Himalayas. Dotted line shows $(pO_2)^{1/2}$ as assumed in some biogeochemical models. **(f)** As **e**, for global pyrite oxidation rate.

The $(pO_2)^{1/2}$ dependency of oxidative weathering on atmospheric oxygen level in the original version of the COPSE model was replaced with the Earth-surface integrated oxidative weathering rate calculated above from the oxidative weathering model, rescaled to give oxidative weathering of $k_{oxidw} = 3.75 \times 10^{12}$ mol C yr$^{-1}$ at pO$_2$ = 1 PAL.

The COPSE land biota (and accompanying weathering enhancement and organic carbon burial) was removed, and solar luminosity was set at 92% of present to generate a steady-state appropriate to 1 Ga, with resulting pCO$_2$ of 26.8 PAL.

A forcing parameter ε was added representing the secular evolution of mechanisms (biological or abiotic) enhancing phosphorus weathering and

bioavailability, and thus controlling marine organic carbon burial. The anoxia dependence of marine organic C/P burial ratio was also switched on (as in run 2 of the original paper[11]).

The sulfur cycle was removed, consistent with the assumption discussed in the main text that sulfur was predominantly in reduced form during the Precambrian.

Atmospheric $CO_2$ was made proportional to the square of the total amount of carbon in the ocean and atmosphere. Long-timescale results of interest here are insensitive to the detailed form of this partitioning.

**Code availability.** The code for the oxidative weathering model and the code for the Precambrian COPSE model (both in MATLAB) are available from S.J.D. on request (S.Daines@exeter.ac.uk). The authors are in the process of developing a new, release version of the COPSE model, details of which are available from the corresponding author (T.M.L.).

**Data availability.** The data that support the findings of this study are the input assumptions of the models (detailed in the 'Methods' section) and the model output (shown in the Figures). These are available from S.J.D. (S.Daines@exeter.ac.uk) or the corresponding author (T.M.L.) on request.

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

## Acknowledgements

This work was supported by the Leverhulme Trust (RPG-2013-106) and the Natural Environment Research Council (NE/N018508/1). We thank Andrey Bekker, Jim Kasting and Tom Laakso for thorough reviews that helped improve the paper.

## Author contributions

S.J.D. and T.M.L. designed the study. S.J.D., B.J.W.M. and T.M.L. developed the model. S.J.D. performed the simulations. T.M.L. and S.J.D. wrote the paper with input from B.J.W.M.

## Additional information

**Competing financial interests:** The authors declare no competing financial interests.

