## [Peer Review File · Nature Communications]

Reviewers' comments:

Reviewer #1 (Remarks to the Author):

Review of the manuscript by Daines et al. titled 'Atmospheric oxygen regulation at low Proterozoic levels by incomplete oxidative weathering of sedimentary organic carbon'

The manuscript provides an innovative modeling of C cycle in the Precambrian based on relationship between pO_2 and organic carbon weathering from sedimentary rocks. This is an entirely new approach that was never before applied to the Precambrian. Modeling suggests relationship between pO_2 and organic carbon weathering from sedimentary rocks (even after the Great Oxidation Event) acting as a stabilizing mechanism for pO_2 and defining fields of long-term stability for pO_2 . It further challenges a recently suggested view of very low oxygen levels in the mid-Proterozoic based on Cr isotope studies. As for any modeling, it is dependent on input parameters and I made detailed comments below. However, I felt that authors carefully addressed possible complications to derive their conclusions. There is underlying assumption that most of organic carbon production is photosynthetic, which I believe authors can easily address and should address in the revised version. In summary, I think this is an excellent study that with minimal revisions should be published in Nature Geoscience. I encourage authors to tighten up text – some of it is partly repetitive and also to justify preferred parameters with references and explanations. Most of it might go to supplementary materials and some of it might be already in tables. If so, it could be referenced in the text more often. In summary, very intriguing work that will likely have huge influence on our understanding how carbon cycle worked in the past.

Detailed comments:

Instead of or in addition to 3, it is best to use the following reference to sulfate evaporites:

Schröder, S., Bekker, A., Beukes, N.J., Strauss, H., van Niekerk, H.S., 2008. Rise in seawater sulphate concentration associated with the Paleoproterozoic positive carbon isotope excursion: evidence from sulphate evaporites in the ~2.2–2.1 Gyr shallow-marine Lucknow Formation, South Africa. *Terra Nova*, 20: 108-117.

Reference 2 was not first to suggest deep-ocean oxygenation during the Lomagundi excursion, see Scott et al., 2008.

Scott, C., Lyons, T.W., Bekker, A., Shen, Y., Poulton, S., Chu, X., Anbar, A., 2008. Tracing the stepwise oxygenation of the Proterozoic biosphere. *Nature*, 452: 456-459.

Lines 33-35: This low level of pO_2 was also challenged by Cr isotope data on carbonates, see Gilleaudeau et al., 2016 *Geochemical Perspectives Letters*.

Great Oxidation – should it be Great Oxidation Event?

I suggest to rephrase the following sentence. It suggests that oxidation is incomplete but it does not imply that it is sensitive to oxygen level. Even if oxygen level would be 10 times pO_2 , probably similar or smaller fraction of Corg would be reburied in high erosion and deposition rate areas. However, at high erosion rates, oxidative weathering is incomplete and detrital kerogen is preserved in marine sediments²⁹⁻³¹, meaning that the global oxidative weathering flux is sensitive to pO_2 variations even at the present day^{26,27}.

Lines 68-70: What do you mean? Why they should even be consistent? Do you imply that volatiles produced during metamorphism are released by volcanic activity? Why? Metamorphic gases are probably released along shear zones and other faults. Should be some volatiles coming from the mantle with volcanic eruptions for sure.

Lines 73-77: What about oxidation of sulfides in the crust and sedimentary rocks? And Fe in the same reservoir? If they are not major players, just explain it.

Lines 78-85: The idea of 'recycled' vs 'new' organic carbon was expressed in Bekker and Holland (2012) and further developed in Bekker (2014) (attached), chapter in Encyclopedia of Astrobiology. These publications should be referenced here.

Lines 89-91: I agree with this general statement, but as you phrase it, it is only one possibility. Other option is that oxygen source was large and oxidative weathering was not kinetically limited. Indeed, detrital pyrite and uraninite are lacking in fluvial successions deposited after 2.4 Ga. Sulfate level increased in seawater.

Lines 130-134: reference here would be useful. Do you imply that 50% of total organic carbon burial happens on land now? Why do you keep Proterozoic marine organic carbon burial at 50% of modern total organic carbon burial if there was no burial of organic carbon on land? With higher nutrient flux from land and low ocean redox state, would not you expect higher marine burial of organic carbon?

Line 147: What do you mean by 'continuous'? Longer-lasting? Extended?

Lines 157-158: What is 'oxygen regulation timescale'? Residence time?

Lines 557-564: This is unclear. What is 38% based on (it seems to be very large number)? What does it mean 'short-term response at constant methane flux'? Are you implying response to perturbation of methane flux or something else? How does flux correspond to burial rates? Via 38%? How did you get these burial rates? I understand 8.1×10^{12} mol C/yr already includes atmospheric methane flux; does it mean you applied 38% twice (what is the difference between black and blue lines)?

The other lines show the short-timescale response at constant methane flux corresponding to burial rates of 8.1×10^{12} mol C yr⁻¹ (modern; blue), 4.0×10^{12} mol C yr⁻¹ (Proterozoic; green), and 2.0×10^{12} mol C yr⁻¹ (Great Oxidation transition; red).

Line 174: I think you mean here dissolved input flux. Total dissolved and solid C weathered fluxes should not change. See Bekker 2014.

Lines 186-188: Reference should be given to Bekker 2014.

Lines 189: add 'new' – changes in new organic carbon burial. Changes in total (=new + recycled) will be still recorded.

Line 192: What do you mean 'until the Phanerozoic'? Is there secular (I presume you mean continuous) trend in the Phanerozoic? Values still fluctuate around 0 permil. Also applies to 252-254.

Lines 192-194: This last sentence is confusing. When erosion rate did not change? In Phanerozoic? What about Wilson cycle? What about large-scale pO₂ changes in the Phanerozoic? This should be clarified.

Line 203: Not sure 'reduce' is right word here. Perturb, affect?

Line 207 – 40 is not appropriate reference for episodic crustal growth

Lines 205-211 – This is likely not correct. Fe oxidation in shales, reflecting average continental crust only changes after the GOE indicating that Fe was not significantly oxidized in the crust

before that. See Bekker and Holland, 2012 for discussion.

Line 210: What is this number based on? I would expect high rate of Fe oxidation in the crust during the GOE, after ~2.32 Ga.

Fig. 2b why similar relationship would not be observed for pyrite?

Lines 212-214: Maybe I missed but I do not remember discussion why 50% of present marine organic carbon burial is so critical. This should be better explained for a uneducated reader. I presume you refer here to Fig. 3b, where 50% is close to the change in slope of Proterozoic (green) curve (please show it with additional line). To the left, at lower organic carbon burial, minor change in organic carbon burial could result in dramatic change in pO₂. Is this what you imply?

Lines 223-224: Not sure why it is kinetically-limited. I would say it is O₂ concentration limited.

Line 231: comma is missing before 'which'

Lines 230-234: What is evidence that oxidized sulfur was immediately reduced? We do not see fluvial or deltaic setting enriched in pyrite, oxidized sulfur certainly made to shelves and into open-ocean.

Lines 237-238: The top of weathering profile was still oxidized. You might want to express it better.

Lines 244-246: There is evidence for survival of detrital pyrite in modern fans around Himalayas. See Maynard et al., 1991.

Lines 252-263: This is partially true, it does not reflect burial of 'new' organic matter, but δ¹³C_{carb} reflect fluctuations in burial of 'new' + 'recycled' organic matter in weakly oxygenated atmosphere.

Line 267: Siderite is very minor mineral in sedimentary sequences.

Line 321, 324: replace 'paper' with 'text' or refer to the paper

Line 323: Parameterize – spelling

Lines 513-515: Not sure if this is correct. Graphite and sulfides would be present in metamorphic rocks of all grades.

Lines 521-522: Is this true? Give the reference. I understand most sulfate precipitate on MOR flanks before it reach feeding zone. What about oxygen?

Lines 526-528: You might want to check old paper by Cameron and Garrels (1980? If my memory is correct). Maybe Chemical Geology? For C:S ratio in Precambrian shales.

Fig. 6. It seems something is not shown correctly in this figure, unless I am not reading it correctly. Why oxidative weathering depth increase with TOC content? Should be opposite relationship.

Andrey Bekker

Reviewer #2 (Remarks to the Author):

The thesis of this paper is that atmospheric O₂ was regulated at low (but not too low) levels during the Proterozoic by a negative feedback involving oxidation of terrestrial organic matter. The authors argue, convincingly, that a large reservoir of organic matter could have built up on the continents during the Archean when atmospheric O₂ levels were low and oxidative weathering did not occur. Following the Great Oxidation Event (GOE), this reduced carbon would have been oxidatively weathered, as it is today. However, if pO₂ was lower than today, then the rate of oxidative weathering may have been O₂-limited. Today, by contrast, oxidative weathering is virtually complete in low-to-moderate erosion areas and is therefore limited by other factors.

Importantly, this eliminates a long-standing problem surrounding the carbon isotope record. Except for brief, but spectacular excursions, the $\delta^{13}\text{C}$ values of marine carbonates have remained roughly constant at ~ 0 permil. For many years, geochemists (including this reviewer) have taken this to mean that the ratio of buried organic carbon to carbonate carbon has remained roughly constant at ~ 0.2 . But the authors demonstrate, again convincingly, that this does not have to be true in an O₂-limited weathering regime, because an increase in organic carbon burial (which removes light carbon preferentially) causes an increase in atmospheric O₂, which in turn causes an increase in oxidative weathering (which puts light carbon back into the atmosphere-ocean system). Brilliant! Why didn't I see this myself years ago? Maybe because I have not spent enough time thinking about this system, and the present authors clearly have spent that time and have thought about this problem more clearly than I have.

The paper has another important implication: The proposed model is consistent with O₂ being regulated at a modestly low level (~ 0.1 times the Present Atmospheric Level, or PAL), but is inconsistent with O₂ regulation at the very low level of 0.001 PAL favored by Tim Lyons, Noah Planavsky, and their colleagues. This is part of an ongoing argument that is well referenced in the present paper. (The opponents are Zhang et al., refs. 16 and 18.) The present paper adds to the argument that Planavsky et al. are wrong and that Proterozoic O₂ was much higher than they estimate.

Usually, I come up with a list of critical comments, but in this case I have only one: Ref. 11 (another paper by coauthor Lenton) does not have the full citation in the reference list. I found it in PNAS, though, and it looks like another strong paper. I'd be happy to see the present paper published essentially as is. Thank you to the authors for clearing up some long-standing misconceptions about the carbon isotope record and for proposing a very believable hypothesis for Proterozoic O₂ regulation. Oh, and I forgot to say: This also makes explaining the GOE much easier because it allows for the possibility of an increase in organic carbon burial as the driver (as the authors point out explicitly in their discussion). So, this model may help to resolve yet another long-standing problem. It deserves to be published quickly.

You may reveal my name to the authors.

Jim Kasting

Reviewer #3 (Remarks to the Author):

Overall comments:

This study uses a detailed model of subaerial weathering, in conjunction with the COPSE model of global biogeochemical cycling, to show that oxygen-sensitive weathering can stabilize pO₂ at 10% PAL, given a lower-than-modern rate of organic carbon burial. The use of a more rigorous weathering model is an important step forward for global biogeochemical modeling, as existing models typically rely on much simpler, poorly-constrained parameterizations.

The paper's dynamical finding -- that a combination of pO₂-sensitive oxidation kinetics and low organic carbon burial allow for stable, Proterozoic levels of atmospheric oxygen -- is similar to a result published by this reviewer (Laakso & Schrag 2014). Though the fundamental conclusions are similar, there are important differences between the two papers; our study used a simplified weathering parameterization, and also called on a different mechanism for reducing organic carbon burial.

A manuscript more focused on the details of novel weathering model, and its relationship to existing parameterizations, would be an excellent contribution to the literature. The results could be even more compelling if used to examine feedbacks that might arise if the organic carbon content of uplifted sediments were explicitly linked to burial rates; see the comments below for lines 95-104. The discussion of the $\delta^{13}\text{C}$ record also addresses an important problem in the field, though some clarification is called for, as discussed in the comments for line 172 and line 254.

Specific comments:

Line 33. It may be worth noting that, though the Proterozoic pO₂ estimate of Planavsky et al. (2014) falls outside the "traditional" range of 1-10% PAL, those bounds are based on poorly constrained models of, for example, paleosol drainage (Rye & Holland 1998) and pyrite oxidation (Canfield & Teske 1996).

Line 71. Figure 1 would benefit from additional labels. It is difficult to infer from the text which fluxes represent detrital transport and deposition versus remineralization. The blue circles seem to denote points of negative feedback, but this is not clearly articulated. Finally, please clarify if this is a schematic or represents the actual fluxes in different runs of the COPSE model.

Lines 78-85. The redox balance being described here is difficult to follow. Given the low rate of oxidative weathering, how is oxygen maintained at low levels in the Archean or the Proterozoic? The negative feedback described in the Abstract suggests this is impossible without some additional dynamical component at work in the global redox, as cycle. Later in the paper, it is mentioned that this is due to a reduction in terrestrial organic carbon burial in your model, but this should be made explicitly clear in this section.

Lines 86-94. This paragraph suggests that the Proterozoic can be distinguished from the Archean by the importance of oxidative weathering kinetics as a source of negative feedback, given the higher level of atmospheric pO₂. I understand this logic, but it is not clear from the text why higher oxygen levels are achieved to begin with, that is, what drives the "secular evolution from net reduced to net oxidized atmospheric inputs". This is certainly a controversial topic, and I realize it is beyond the scope of this paper to answer the question of what caused GOE. However, it would be beneficial to cite some of the many proposals, if the paper is going to include a discussion of the differences in Archean and Proterozoic redox cycling.

Lines 95-104. This is a novel approach, and a welcome development for the field; many of the existing global redox models have included simplistic parameterizations of oxidative weathering, often ultimately based on Chang & Berner's (1999) square-root O₂ dependency. It is reassuring that this functional form is not a terrible fit to the results shown in Figure 2a! This should be discussed briefly in the paper's main text as well. Given that this model takes explicit account of organic carbon, it presents a great opportunity to study an often-neglected feedback: how does the reduced organic carbon burial required to simulate the Proterozoic feedback on organic carbon weathering rates following uplift?

Lines 117-119. This sentence should be rephrased for clarity.

Lines 125-127. I agree that atmospheric oxidation rates saturate at Proterozoic oxygen levels, but

a citation, or an explanation rooted in the model, should be provided. (In other words, where does the “volcanic” term in Figure 2b come from?) Aqueous oxidation of sulfide and ferrous iron may also be important sinks to compare to, given that the ocean was likely ferruginous through most of the Proterozoic (e.g., Canfield & Poulton 2011).

Lines 130-133. It is interesting that the weathering feedback can stabilize pO₂ at Proterozoic levels given reduced organic carbon burial, but this has been shown previously (Laakso & Schrag 2014), albeit using a simple square root dependence on pO₂ for the weathering rate.

A discussion of why this lower burial rate can be assumed would also be helpful for the reader – perhaps a review of the results of Lenton et al. (2016)?

Lines 151-161. The dynamic perspective described here, in which the source of reductants to the surface environment is a function of marine burial rates, has been lacking in many previous models. The weathering model described here presents a great opportunity to examine this feedback in more detail. My reading of the method section suggests organic carbon content of uplifted material was maintained at modern levels. (Though pyrite content was effectively increased by fixing the C:S ratio while allowing all sulfur to be present as pyrite.) Given that less organic carbon is being buried in the Proterozoic model, should we expect the global rates of oxidative weathering to slow down as well, even at a given oxygen level? This would be a very interesting test to run.

Lines 162-171. Though it is cited earlier in the manuscript, Claire et al. (2006) should be cited here, as they have examined the effect of falling outgassing rates and changing crustal composition in some quantitative detail.

Line 172 (and following section). This section seems to argue that sedimentary $\delta^{13}\text{C}$ should remain stable over most of Earth history because net burial of organic carbon must remain constant to balance both the oxygen and carbon budgets. This implies that the “f-ratio”, which is measurement of net burial (i.e burial minus weathering; this is why is traditionally balanced with only volcanic inputs), does indeed remain constant over Earth history, with the model results providing a mechanism for explaining that stability. It is unclear why a contrast has been drawn with previous discussions of the f-ratio (both here and at line 254), which have observed this constancy but not explained it. Is there a disagreement here over definition?

Some discussion of the sulfur cycle is called for. Specifically, it would be useful to understand why, when oxygen levels are forced to another steady state (as in Figure 3), the model compensates for changes in organic carbon burial entirely with changes in organic carbon weathering (allowing fixed $\delta^{13}\text{C}_{\text{carb}}$), when changes in pyrite weathering and burial should also have their own, possibly complex response to pO₂.

Lines 200-204. Both the Holland (2002) and Kump & Barley (2007) argue that these processes would have led to lower rates of reductant input after the GOE, not higher.

Lines 212-218. See comment on Lines 78-85. A more detailed discussion of elimination of the terrestrial biosphere, and its effects on both NPP and subsequent remineralization and nutrient mobilization processes, would be helpful.

Line 224. The model presented here stabilizes at 0.1 PAL, and a clear dynamical explanation is given. However, are the parameterizations robust enough to distinguish between stability at 0.1 and 0.001 PAL, especially given the arbitrary nature of the chosen organic carbon burial rate? The argument provided at lines 237-251 is more grounded in well-constrained observations (i.e., Johnson et al. 2014), though it should be noted that this study did not conclude pO₂ was necessarily as high as 10% after the GOE.

Lines 252-263. Please see comment for line 172.

Lines 274-284. This is an important paragraph, and should be discussed earlier in the manuscript. Maintaining low organic carbon burial is key to their model, but the specified burial rate (50% of modern) has not been clearly explained as function of land plant evolution. The cited model (Lenton et al. 2016) does show that the appearance of land plants can lead to increased oxygen levels in the COPSE model, but their pre-Devonian pO₂ value is of order 20% PAL, substantially higher than in the model here, and many redox proxies should show a systematic difference between the early Paleozoic and the Proterozoic (see Lyons et al. 2014 for a summary). Though a factor of 2 is potentially justifiable, this important factor requires further discussion, and should be articulated clearly earlier in the paper.

--Thomas Laakso

References:

- Lenton et al. (2016) PNAS 113 9704.
- Planavsky et al. (2014) Science 346 635.
- Canfield & Teske (1996) Nature 382 127.
- Rye & Holland (1998) Am. J. Sci. 298 621.
- Laakso & Schrag (2014) EPSL 388 81
- Lyons et al. (2014) 506 307
- Chang & Berner (1999) GCA 63 3301
- Canfield & Poulton (2011) Elements 7 107
- Claire et al. (2006) Geobiology 4 239
- Holland (2002) GCA 66 3811
- Kump & Barley (2007) 448 1033
- Johnson et al. (2014) GSA Bull. 126 813

Response to Reviewers' comments

Reviewers' comments are in black.

Our responses are in red.

The revised version of the paper has the text changes highlighted for ease of reference.

Reviewer #1 – Andrey Bekker

The manuscript provides an innovative modeling of C cycle in the Precambrian based on relationship between pO₂ and organic carbon weathering from sedimentary rocks. This is an entirely new approach that was never before applied to the Precambrian. Modeling suggests relationship between pO₂ and organic carbon weathering from sedimentary rocks (even after the Great Oxidation Event) acting as a stabilizing mechanism for pO₂ and defining fields of long-term stability for pO₂. It further challenges a recently suggested view of very low oxygen levels in the mid-Proterozoic based on Cr isotope studies. As for any modeling, it is dependent on input parameters and I made detailed comments below. However, I felt that authors carefully addressed possible complications to derive their conclusions.

There is underlying assumption that most of organic carbon production is photosynthetic, which I believe authors can easily address and should address in the revised version.

We have added a sentence when first discussing our Proterozoic scenario in the Results: “We assume by this time the majority of organic carbon was produced by oxygenic photosynthesis.”

In summary, I think this is an excellent study that with minimal revisions should be published in Nature Geoscience.

We note that the paper is under consideration at Nature Communications because the Nature Geoscience editors declined to send it out to review.

I encourage authors to tighten up text – some of it is partly repetitive and also to justify preferred parameters with references and explanations. Most of it might go to supplementary materials and some of it might be already in tables. If so, it could be referenced in the text more often.

We have sought to tighten up the text and make more references to the Tables and Methods to justify parameter choices. We have resisted creating a supplementary information in the interests of having a self-contained paper.

In summary, very intriguing work that will likely have huge influence on our understanding how carbon cycle worked in the past.

We thank Andrey for their very positive response to the paper

Detailed comments:

Instead of or in addition to 3, it is best to use the following reference to sulfate evaporites:

Schröder, S., Bekker, A., Beukes, N.J., Strauss, H., van Niekerk, H.S., 2008. Rise in seawater sulphate concentration associated with the Paleoproterozoic positive carbon isotope excursion: evidence from sulphate evaporites in the ~2.2–2.1 Gyr shallow-marine Lucknow Formation, South Africa. *Terra Nova*, 20: 108-117.

Thanks – this has now been referenced in addition.

Reference 2 was not first to suggest deep-ocean oxygenation during the Lomagundi excursion, see Scott et al., 2008.

Scott, C., Lyons, T.W., Bekker, A., Shen, Y., Poulton, S., Chu, X., Anbar, A., 2008. Tracing the stepwise oxygenation of the Proterozoic biosphere. *Nature*, 452: 456-459.

Apologies for this omission, this reference has now been added.

Lines 33-35: This low level of pO₂ was also challenged by Cr isotope data on carbonates, see Gilleaudeau et al., 2016 *Geochemical Perspectives Letters*.

We have added reference to this new paper “...and work showing chromium isotope fractionation in carbonates ~1.1 Ga onwards²¹”. We have also clarified that the lack of chromium isotope fractionation found by Planavsky et al. is “in iron-rich sedimentary rocks¹⁴”.

Great Oxidation – should it be Great Oxidation Event?

We have changed to Great Oxidation Event throughout.

I suggest to rephrase the following sentence. It suggests that oxidation is incomplete but it does not imply that it is sensitive to oxygen level. Even if oxygen level would be 10 times pO₂, probably similar or smaller fraction of Corg would be reburied in high erosion and deposition rate areas.

However, at high erosion rates, oxidative weathering is incomplete and detrital kerogen is preserved in marine sediments²⁹⁻³¹, meaning that the global oxidative weathering flux is sensitive to pO₂ variations even at the present day^{26,27}.

There is a causal relation here – if oxidative weathering is incomplete because erosion takes Corg out of contact with O₂ before it can be fully oxidised, then increasing pO₂ will lead to a greater consumption of Corg (i.e. a smaller fraction of Corg will be reburied). We acknowledge that the oxidation may remain incomplete in fast eroding environments up to much higher pO₂ levels. We have added a “somewhat” sensitive to the sentence to cover this.

Lines 68-70: What do you mean? Why they should even be consistent? Do you imply that volatiles produced during metamorphism are released by volcanic activity? Why? Metamorphic gases are probably released along shear zones and other faults. Should be some volatiles coming from the mantle with volcanic eruptions for sure.

We have removed the offending sentence from the main text, as it is tangential to the main argument here. We meant, as stated in the table footnote, that: “The reasonable agreement of

these totals is consistent with the volcanic reduced gases coming predominantly from high temperature metamorphism of sediments (rather than the mantle) and therefore these are two estimates of the same oxygen sink.” However, we agree that metamorphic volatiles are not necessarily released through volcanic activity, and that there are some volcanic volatiles coming from the mantle.

Lines 73-77: What about oxidation of sulfides in the crust and sedimentary rocks? And Fe in the same reservoir? If they are not major players, just explain it.

Our focus is on organic carbon. Some observations about the size of the oxidative sinks of sulfur and iron were in parentheses at the end of the previous paragraph, but we have deleted the content there and added a sentence here that: “Oxidation of sulfides and ferrous iron in sediments and the upper crust are smaller sinks at present, because less than half of the sedimentary iron and sulfur are in reduced form.”

Lines 78-85: The idea of ‘recycled’ vs ‘new’ organic carbon was expressed in Bekker and Holland (2012) and further developed in Bekker (2014) (attached), chapter in Encyclopedia of Astrobiology. These publications should be referenced here.

We have added references to both of these papers at the end of this paragraph.

Lines 89-91: I agree with this general statement, but as you phrase it, it is only one possibility. Other option is that oxygen source was large and oxidative weathering was not kinetically limited. Indeed, detrital pyrite and uraninite are lacking in fluvial successions deposited after 2.4 Ga. Sulfate level increased in seawater.

The oxidation of pyrite and uraninite is currently understood to be much faster than that of kerogen hence they can be completely oxidised at pO_2 levels where kerogen is still incompletely oxidised (as predicted for pyrite in the model used herein). It is conceivable that the oxygen source increased a lot, in particular during the transient Lomagundi event, in which case the oxidative weathering sink might have been temporarily overwhelmed, but to cover this possibility at this point would disrupt the flow of the argument. Instead we have added the following to the Discussion: “We have assumed that any increase in oxygen source triggering the Great Oxidation Event was modest, but if the oxygen source increased a lot, e.g. in the Lomagundi event, then the oxidative weathering sink might have been temporarily overwhelmed leading to high oxygen levels²⁻⁷. Either way by ~1.85 Ga oxygen levels were low again.”

Lines 130-134: reference here would be useful. Do you imply that 50% of total organic carbon burial happens on land now? Why do you keep Proterozoic marine organic carbon burial at 50% of modern total organic carbon burial if there was no burial of organic carbon on land? With higher nutrient flux from land and low ocean redox state, would not you expect higher marine burial of organic carbon?

We imply that around 50% of today’s total organic carbon burial is derived from land plants, but not that it is buried on land – rather, most of it is eroded to deltaic and shelf sea settings and buried there. We have added references to Kump 1988, Bergman et al. 2004 and Lenton et al. 2016 for the ~50% figure (and added a “~” to indicate it is a rough fraction). The key point about this terrestrially-derived matter is that it has a much higher C/P>1000 than marine-derived organic matter (C/P~100), therefore transferring the corresponding P to the

ocean does not give rise to a corresponding increase in marine organic carbon burial. To clarify our assumptions we have expanded this section: “Assuming negligible burial of terrestrial organic matter in the Proterozoic, which comprises ~50% of total burial today^{11,13,45}, and assuming modern ocean nutrient levels, marine organic carbon burial could have been comparable to the modern level of $\sim 2.5 \times 10^{12}$ mol C yr⁻¹. Combining this with minimum estimates for volcanic and metamorphic inputs, the model predicts $pO_2 \sim 0.1$ PAL (Fig. 3a) with oxygen stabilized by the negative feedback from oxidative weathering.”

Line 147: What do you mean by ‘continuous’? Longer-lasting? Extended?

We mean that there is no mathematical discontinuity in the steady state solutions for atmospheric oxygen, as there is at the Great Oxidation Event. We have replaced “more continuous” with “not as abrupt” in the hope this is clearer.

Lines 157-158: What is ‘oxygen regulation timescale’? Residence time?

They are comparable in this case, but technically the regulation timescale would be the e-folding timescale of recovery from a perturbation. As they are comparable we have replaced “regulation timescale” with “residence time”.

Lines 557-564: This is unclear. What is 38% based on (it seems to be very large number)? What does it mean ‘short-term response at constant methane flux’? Are you implying response to perturbation of methane flux or something else? How does flux correspond to burial rates? Via 38%? How did you get these burial rates? I understand 8.1×10^{12} mol C/yr already includes atmospheric methane flux; does it mean you applied 38% twice (what is the difference between black and blue lines)?

The 38% is indeed an alarmingly high number – the 1.55×10^{12} mol CH₄/yr value is referenced in Table 1 and caveats with it are noted in a footnote there. ‘Short term response at constant methane flux’ means we assume that the methane flux is constant at 38% of the stated initial burial rates, rather than scaling with the organic carbon burial flux (e.g. corresponding to O₂ consumption of 3.1×10^{12} mol O₂/yr throughout in the case of the blue line). This is referred to as the ‘short term response’ because we presume that on long timescales the metamorphic flux of methane must be proportional to the organic carbon flux previously deposited. The burial rate of 2.0×10^{12} mol O₂/yr corresponds to the Great Oxidation because this just balances 1.25×10^{12} mol O₂/yr volcanic gases + $(0.38 \times 2.0 = 0.76) \times 10^{12}$ mol O₂/yr metamorphic methane (with no oxidative weathering). The burial rate of 4.0×10^{12} mol O₂/yr is illustrative and is balanced by 1.25×10^{12} mol O₂/yr volcanic gases + $(0.38 \times 4.0 = 1.52) \times 10^{12}$ mol O₂/yr metamorphic methane + (variable) oxidative weathering flux. The blue line is the case with 1.25×10^{12} mol O₂/yr fixed volcanic gas sink + $(0.38 \times 8.1 = 3.078) \times 10^{12}$ mol O₂/yr fixed metamorphic methane sink + a variable oxidative weathering sink, whereas the black line is the case with 1.25×10^{12} mol O₂/yr fixed volcanic gas sink + variable metamorphic methane sink (always 38% of organic carbon burial) + variable oxidative weathering sink. Thus we do not apply 38% twice. We have clarified the figure caption accordingly: “The other lines show the response with a constant metamorphic methane flux, fixed at 38% of initial burial rates of 8.1×10^{12} mol C yr⁻¹ (modern; blue), 4.0×10^{12} mol C yr⁻¹ (Proterozoic; green), and 2.0×10^{12} mol C yr⁻¹ (Great Oxidation Event transition; red). These cases represent the short-term response to perturbations because we presume that on long timescales the metamorphic flux of methane must be proportional to the organic carbon flux previously deposited.”

Line 174: I think you mean here dissolved input flux. Total dissolved and solid C weathered fluxes should not change. See Bekker 2014.

As Jim Kasting's review confirms and the papers we cite indicate, many prominent studies have interpreted the constancy of $\delta^{13}\text{C}_{\text{carb}}$ as implying a constant ratio of 'new' organic to inorganic carbon burial. We recognise that Andrey's thinking has advanced beyond this.

Lines 186-188: Reference should be given to Bekker 2014.

This has been added.

Lines 189: add 'new' – changes in new organic carbon burial. Changes in total (=new + recycled) will be still recorded.

Agreed, we have added "new".

Line 192: What do you mean 'until the Phanerozoic'? Is there secular (I presume you mean continuous) trend in the Phanerozoic? Values still fluctuate around 0 permil. Also applies to 252-254.

Secular trend means a long-term (in this case positive) trend. In the Phanerozoic there is a shift from 0 to 2 per mil associated with the rise of land plants, as explored in Lenton et al. 2016. Since around 445 Ma values have fluctuated around 2 per mil rather than 0 per mil, based on taking the long-term running mean of the $\delta^{13}\text{C}_{\text{carb}}$ compilation by Saltzman and Thomas in the Geological Time Scale 2012. This has been clarified by adding: "(where there is a shift from 0 to 2‰ associated with the rise of land plants¹³)"

Lines 192-194: This last sentence is confusing. When erosion rate did not change? In Phanerozoic? What about Wilson cycle? What about large-scale pO_2 changes in the Phanerozoic? This should be clarified.

We do not mean to imply that erosion rates haven't changed rather this is an idealised scenario that is being described. It has been clarified with: "As long as the erosion rate is unchanged..."

Line 203: Not sure 'reduce' is right word here. Perturb, affect?

We have changed it to "affect".

Line 207 – 40 is not appropriate reference for episodic crustal growth

We have replaced this reference with citation of the original proposal of episodic continental growth in this interval by McLennan and Taylor (1982).

Lines 205-211 – This is likely not correct. Fe oxidation in shales, reflecting average continental crust only changes after the GOE indicating that Fe was not significantly oxidized in the crust before that. See Bekker and Holland, 2012 for discussion.

This is a good point. We have changed “implies” to “would imply” and “drops” to “would drop” to indicate the hypothetical nature of the scenario described. We have added a sentence at the end of the paragraph citing Bekker and Holland: “However, the iron oxidation state of shales only changes after the Great Oxidation Event, suggesting that iron was not significantly oxidized in the crust beforehand⁶.”

Line 210: What is this number based on? I would expect high rate of Fe oxidation in the crust during the GOE, after ~2.32 Ga.

The number is based on oxidising the remaining 20% of 2×10^{21} molO₂eq that is created over the following 2.5×10^9 yr = 1.6×10^{11} molO₂eq/yr. The question is really whether sufficient oxidant was available pre-GOE to do the oxidation of new crust then or whether it had to await the GOE. Either way, as we note earlier in the paragraph, the bulk of crustal iron oxidation is expected to be complete by the end of the Lomagundi excursion.

Fig. 2b why similar relationship would not be observed for pyrite?

Pyrite oxidation kinetics are faster hence the sensitivity of this sink to pO₂ is at lower pO₂. Crucially this sink is not part of Proterozoic oxygen regulation – as we explain in the Discussion – because the sedimentary sulphur reservoir was largely in reduced form.

Lines 212-214: Maybe I missed but I do not remember discussion why 50% of present marine organic carbon burial is so critical. This should be better explained for a uneducated reader. I presume you refer here to Fig. 3b, where 50% is close to the change in slope of Proterozoic (green) curve (please show it with additional line). To the left, at lower organic carbon burial, minor change in organic carbon burial could result in dramatic change in pO₂. Is this what you imply?

The crucial figure is 25% of present total organic carbon burial, because this corresponds to the oxygen-insensitive sink of O₂ by reaction with volcanic reduced gases. Below this level of organic carbon burial a dramatic drop in pO₂ occurs until the oxidation of reduced volcanic gases becomes kinetically sensitive to pO₂ (i.e. a reducing atmosphere). In response also to Thomas Laakso’s review we have adjusted this section: “Our proposed Proterozoic oxygen regulation mechanism requires that marine organic carbon burial remained at least ~25% of today’s total organic carbon burial flux after the Great Oxidation Event. Marine-derived organic carbon is estimated to comprise half^{11,13} to two-thirds⁵⁴ of today’s total organic carbon burial flux (the remainder being derived from land plants). Hence marine organic carbon burial must have remained at least ~40-50% of its present value since the Great Oxidation Event.”

Lines 223-224: Not sure why it is kinetically-limited. I would say it is O₂ concentration limited.

They mean the same thing, except kinetically-limited is more general (allowing that either reactant concentration may determine the reaction rate). Nevertheless we have added “(i.e. oxygen concentration limited)”.

Line 231: comma is missing before ‘which’

Corrected.

Lines 230-234: What is evidence that oxidized sulfur was immediately reduced? We do not see fluvial or deltaic setting enriched in pyrite, oxidized sulfur certainly made to shelves and into open-ocean.

We have changed “immediately” to “soon” – the point is that before returning to the sedimentary reservoir the great majority of the sulfur was reduced.

Lines 237-238: The top of weathering profile was still oxidized. You might want to express it better.

We have added “beneath the upper soil layers...”

Lines 244-246: There is evidence for survival of detrital pyrite in modern fans around Himalayas. See Maynard et al., 1991.

We have added a sentence and the suggested reference: “Indeed some detrital pyrite survives oxidation today in the most rapidly eroding settings⁵⁵.”

Lines 252-263: This is partially true, it does not reflect burial of ‘new’ organic matter, but $\delta^{13}\text{C}_{\text{carb}}$ reflect fluctuations in burial of ‘new’ + ‘recycled’ organic matter in weakly oxygenated atmosphere.

We agree and have clarified the paragraph by specifying “new” organic carbon burial throughout.

Line 267: Siderite is very minor mineral in sedimentary sequences.

Fair point, but we felt it reasonable to cite the recent work by Bachan and Kump on this.

Line 321, 324: replace ‘paper’ with ‘text’ or refer to the paper

Fixed.

Line 323: Parameterize – spelling

Fixed.

Lines 513-515: Not sure if this is correct. Graphite and sulfides would be present in metamorphic rocks of all grades.

We agree and have clarified that this is an “Upper” estimate, adding at the end: “(i.e. neglecting graphite carbon and sulfides in metamorphic rocks)”.

Lines 521-522: Is this true? Give the reference. I understand most sulfate precipitate on MOR flanks before it reach feeding zone. What about oxygen?

Apologies, we had the wrong Sleep (2005) paper referenced here, this has now been corrected and cited in the footnote as it explains that in a low sulfate (and anoxic) ancient deep ocean there would have been little or no seafloor basalt oxidation.

Lines 526-528: You might want to check old paper by Cameron and Garrels (1980? If my memory is correct). Maybe Chemical Geology? For C:S ratio in Precambrian shales.

Thanks, we have checked this paper which gives a C:S ratio of 0.36 for Proterozoic shales, which as they note is similar to that of modern sediments and close to the value we assume.

Fig. 6. It seems something is not shown correctly in this figure, unless I am not reading it correctly. Why oxidative weathering depth increase with TOC content? Should be opposite relationship.

What we are plotting is the TOC remaining as a function of depth for different O₂ concentrations. Thus TOC is generally greater at depths where O₂ cannot penetrate, and the higher the O₂ the deeper the weathering horizon. We have expanded the legend to explain this: "The lines show the TOC remaining as a function of depth for different prescribed pO₂ (PAL) levels. In each case a distinct weathering horizon forms (above which TOC has been oxidized and below which it has not), with the weathering horizon at greater depth for higher pO₂."

Andrey Bekker

We thank Andrey for his very careful and thorough review, and have added an Acknowledgement.

Reviewer #2 – Jim Kasting

The thesis of this paper is that atmospheric O₂ was regulated at low (but not too low) levels during the Proterozoic by a negative feedback involving oxidation of terrestrial organic matter. The authors argue, convincingly, that a large reservoir of organic matter could have built up on the continents during the Archean when atmospheric O₂ levels were low and oxidative weathering did not occur. Following the Great Oxidation Event (GOE), this reduced carbon would have been oxidatively weathered, as it is today. However, if pO₂ was lower than today, then the rate of oxidative weathering may have been O₂-limited. Today, by contrast, oxidative weathering is virtually complete in low-to-moderate erosion areas and is therefore limited by other factors.

Importantly, this eliminates a long-standing problem surrounding the carbon isotope record. Except for brief, but spectacular excursions, the $\delta^{13}\text{C}$ values of marine carbonates have remained roughly constant at ~ 0 permil. For many years, geochemists (including this reviewer) have taken this to mean that the ratio of buried organic carbon to carbonate carbon has remained roughly constant at ~ 0.2 . But the authors demonstrate, again convincingly, that this does not have to be true in an O₂-limited weathering regime, because an increase in organic carbon burial (which removes light carbon preferentially) causes an increase in atmospheric O₂, which in turn causes an increase in oxidative weathering (which puts light carbon back into the atmosphere-ocean system). Brilliant! Why didn't I see this myself years ago? Maybe because I have not spent enough time thinking about this system, and the present authors clearly have spent that time and have thought about this problem more clearly than I have.

The paper has another important implication: The proposed model is consistent with O₂ being regulated at a modestly low level (~0.1 times the Present Atmospheric Level, or PAL), but is inconsistent with O₂ regulation at the very low level of 0.001 PAL favored by Tim Lyons, Noah Planavsky, and their colleagues. This is part of an ongoing argument that is well referenced in the present paper. (The opponents are Zhang et al., refs. 16 and 18.) The present paper adds to the argument that Planavsky et al. are wrong and that Proterozoic O₂ was much higher than they estimate.

Usually, I come up with a list of critical comments, but in this case I have only one: Ref. 11 (another paper by coauthor Lenton) does not have the full citation in the reference list. I found it in PNAS, though, and it looks like another strong paper.

Thanks – the missing reference details have been added for this paper.

I'd be happy to see the present paper published essentially as is. Thank you to the authors for clearing up some long-standing misconceptions about the carbon isotope record and for proposing a very believable hypothesis for Proterozoic O₂ regulation. Oh, and I forgot to say: This also makes explaining the GOE much easier because it allows for the possibility of an increase in organic carbon burial as the driver (as the authors point out explicitly in their discussion). So, this model may help to resolve yet another long-standing problem. It deserves to be published quickly.

You may reveal my name to the authors.

Jim Kasting

We thank Jim for his very positive response to the paper, and have added an Acknowledgement.

Reviewer #3 – Thomas Laakso

Overall comments:

This study uses a detailed model of subaerial weathering, in conjunction with the COPSE model of global biogeochemical cycling, to show that oxygen-sensitive weathering can stabilize pO₂ at 10% PAL, given a lower-than-modern rate of organic carbon burial. The use of a more rigorous weathering model is an important step forward for global biogeochemical modeling, as existing models typically rely on much simpler, poorly-constrained parameterizations.

The paper's dynamical finding -- that a combination of pO₂-sensitive oxidation kinetics and low organic carbon burial allow for stable, Proterozoic levels of atmospheric oxygen -- is similar to a result published by this reviewer (Laakso & Schrag 2014). Though the fundamental conclusions are similar, there are important differences between the two papers; our study used a simplified weathering parameterization, and also called on a different mechanism for reducing organic carbon burial.

Thanks – we recognise the important contribution of the earlier paper by Laakso & Schrag and the fact that your model includes oxygen-sensitive oxidative weathering. However, it

does not consider the cycling of sedimentary reservoirs, which is central to our study. Furthermore, your suggestion of a 10-fold reduction in P flux to the ocean giving rise to $pO_2 \sim 0.1$ PAL would not give the same result in our model because you have neglected the pO_2 -insensitive sinks of oxygen from oxidation of metamorphic reduced gases. Thus the predicted relationship between organic carbon burial and pO_2 is quite different in our study.

A manuscript more focused on the details of novel weathering model, and its relationship to existing parameterizations, would be an excellent contribution to the literature. The results could be even more compelling if used to examine feedbacks that might arise if the organic carbon content of uplifted sediments were explicitly linked to burial rates; see the comments below for lines 95-104.

In the COPSE model the organic carbon content of uplifted sediments is already ultimately linked to burial rates in that changes in burial do change the total sedimentary organic carbon reservoir – but as that is an enormous reservoir the feedback is very slow (on the sedimentary reservoir recycling timescale ~ 150 Myr) and weak. In a sense we already discuss this feedback when we note how a large sedimentary organic carbon reservoir would have accumulated in the Archean in the absence of oxidative weathering, but after the Great Oxidation the organic-rich sediments would then be subject to oxidative weathering. Further comments on the possibility of more ‘rapid recycling’ are given below.

The discussion of the $\delta^{13}C$ record also addresses an important problem in the field, though some clarification is called for, as discussed in the comments for line 172 and line 254.

We have added clarification as described below.

Specific comments:

Line 33. It may be worth noting that, though the Proterozoic pO_2 estimate of Planavsky et al. (2014) falls outside the “traditional” range of 1-10% PAL, those bounds are based on poorly constrained models of, for example, paleosol drainage (Rye & Holland 1998) and pyrite oxidation (Canfield & Teske 1996).

We have cited these studies and elaborated that these estimates are “based on models of paleosol oxidation⁸ and of benthic sulfide oxidation¹⁸ that can be questioned.”

Line 71. Figure 1 would benefit from additional labels. It is difficult to infer from the text which fluxes represent detrital transport and deposition versus remineralization. The blue circles seem to denote points of negative feedback, but this is not clearly articulated. Finally, please clarify if this is a schematic or represents the actual fluxes in different runs of the COPSE model.

We have elaborated the caption of Figure 1, clarifying that it is a schematic, making clear what the blue circles refer to and the nature of the key negative feedback in each case. Detrital transport followed by deposition is indicated by a black looped arrow coming out of the sedimentary carbon reservoir and going back into it without any corresponding oxygen fluxes. This is clarified in the main text as “(indicated by the thin looped black arrow going from and to the sediments in Fig. 1d)”.

Lines 78-85. The redox balance being described here is difficult to follow. Given the low rate

of oxidative weathering, how is oxygen maintained at low levels in the Archean or the Proterozoic? The negative feedback described in the Abstract suggests this is impossible without some additional dynamical component at work in the global redox, as cycle. Later in the paper, it is mentioned that this is due to a reduction in terrestrial organic carbon burial in your model, but this should be made explicitly clear in this section.

The oxygen regulation mechanism (negative feedback) in the Archean scenario is due to the oxygen sensitivity of reaction with atmospheric reductants, which only kicks in at very low pO₂ levels. In this regime variations in organic carbon burial are counterbalanced by variations in the oxidation of metamorphic reduced gases. This has now been clarified in the text by adding: “with oxygen stabilized instead by the oxygen-sensitivity of reactions with reduced gases”

Lines 86-94. This paragraph suggests that the Proterozoic can be distinguished from the Archean by the importance of oxidative weathering kinetics as a source of negative feedback, given the higher level of atmospheric pO₂. I understand this logic, but it is not clear from the text why higher oxygen levels are achieved to begin with, that is, what drives the “secular evolution from net reduced to net oxidized atmospheric inputs”. This is certainly a controversial topic, and I realize it is beyond the scope of this paper to answer the question of what caused GOE. However, it would be beneficial to cite some of the many proposals, if the paper is going to include a discussion of the differences in Archean and Proterozoic redox cycling.

We already cite papers by Kasting (2013), Claire et al. (2006), Zahnle et al. (2013) and Catling (2014) at this point – these represent the most recent and significant quantitative reviews and analyses of the various proposals for the secular evolution triggering the GOE.

Lines 95-104. This is a novel approach, and a welcome development for the field; many of the existing global redox models have included simplistic parameterizations of oxidative weathering, often ultimately based on Chang & Berner’s (1999) square-root O₂ dependency. It is reassuring that this functional form is not a terrible fit to the results shown in Figure 2a! This should be discussed briefly in the paper’s main text as well.

We have added a sentence in the results section including citations of Chang and Berner (1999) and Laakso and Schrag (alongside Bergman et al.): “Overall a dependency of oxidative weathering on pO₂^{0.5} (Fig. 2a, dashed line), consistent with coal oxidation kinetics⁴⁵ and as assumed in some previous biogeochemical models^{11,25}, provides a reasonable fit to the results here.” We have also added citations of Chang and Berner and Laakso and Schrag to the figure caption.

Given that this model takes explicit account of organic carbon, it presents a great opportunity to study an often-neglected feedback: how does the reduced organic carbon burial required to simulate the Proterozoic feedback on organic carbon weathering rates following uplift?

This comment is a little hard to follow. We think it alludes to what Berner described as ‘rapid recycling’ i.e. that changes in burial of organic carbon on continental shelves can sometimes be followed geologically soon after by the exposure of the same sediments (e.g. due to sea-level fall, or indeed uplift) leading to counterbalancing changes in oxidative weathering. This would be an interesting topic for further work, but it won’t fundamentally alter our results because our mechanism hinges on there being a large excess of available organic carbon for

oxidative weathering in the Proterozoic. We already include a sensitivity analysis for varying uplift in that we consider a limiting case where all rapidly eroding terrains are excluded (noting that today is a high uplift world).

Lines 117-119. This sentence should be rephrased for clarity.

We have rephrased this sentence: “As pO_2 declines further, below $pO_2 \sim 0.03$ PAL, although oxidative weathering remains sensitive to pO_2 the potential negative feedback becomes weaker as this sink becomes relatively small (Fig. 2b).”

Lines 125-127. I agree that atmospheric oxidation rates saturate at Proterozoic oxygen levels, but a citation, or an explanation rooted in the model, should be provided. (In other words, where does the “volcanic” term in Figure 2b come from?)

We have added citations to Claire et al. (2006) and Catling (2014) at this point who show the saturation of the oxidation rate of methane (an indicative reduced gas) as pO_2 rises, as well as the oxidation of various reduced gaseous sulfur compounds. The volcanic term in Figure 2b is a schematic parameterisation to show pO_2 sensitivity at pre-GOE ($<10^{-5}$ PAL) levels and saturation at Proterozoic levels, and we now make this clear in the caption, again citing Claire et al. and Catling: “The organic carbon and pyrite results are from the modelling herein, whereas the volcanic reduced gas consumption is a schematic curve to show pO_2 sensitivity at pre-Great Oxidation Event levels ($\leq 10^{-5}$ PAL) but not at Proterozoic levels^{40,42} (hence this process cannot help explain Proterozoic atmospheric oxygen regulation).”

Aqueous oxidation of sulfide and ferrous iron may also be important sinks to compare to, given that the ocean was likely ferruginous through most of the Proterozoic (e.g., Canfield & Poulton 2011).

Aqueous oxidation of sulfide or ferrous iron are not pertinent to Proterozoic oxygen regulation because the Proterozoic sedimentary reservoir of each of these elements was dominated either by the reduced form (for sulfur) or the oxidised form (for iron). This is explained for the case of sulfur in the Discussion.

Lines 130-133. It is interesting that the weathering feedback can stabilize pO_2 at Proterozoic levels given reduced organic carbon burial, but this has been shown previously (Laakso & Schrag 2014), albeit using a simple square root dependence on pO_2 for the weathering rate.

This point is well taken. Indeed this square root relationship is already included in the original COPSE model and is (at least partly) responsible for setting its early Paleozoic oxygen level. The novelty in the present study is meant to be the more complete consideration of sedimentary cycling over Earth history, the more detailed modelling of the oxidative weathering feedback, and the implications of all this for interpreting the $\delta^{13}C$ record.

A discussion of why this lower burial rate can be assumed would also be helpful for the reader – perhaps a review of the results of Lenton et al. (2016)?

We now cite Kump 1988, Bergman et al. 2004, and Lenton et al. 2016 in relation to the ~50% contribution of terrestrial-derived material to present total organic carbon burial. Furthermore we have expanded this section to explain our assumptions: “Assuming negligible burial of

terrestrial organic matter in the Proterozoic, which comprises ~50% of total burial today^{11,13,45}, and assuming modern ocean nutrient levels, marine organic carbon burial could have been comparable to the modern level of $\sim 2.5 \times 10^{12}$ molC yr⁻¹.” We also return to this issue with an expanded treatment in the Discussion.

Lines 151-161. The dynamic perspective described here, in which the source of reductants to the surface environment is a function of marine burial rates, has been lacking in many previous models. The weathering model described here presents a great opportunity to examine this feedback in more detail. My reading of the method section suggests organic carbon content of uplifted material was maintained at modern levels. (Though pyrite content was effectively increased by fixing the C:S ratio while allowing all sulfur to be present as pyrite.) Given that less organic carbon is being buried in the Proterozoic model, should we expect the global rates of oxidative weathering to slow down as well, even at a given oxygen level? This would be a very interesting test to run.

The COPSE model dynamically simulates the sedimentary organic carbon reservoir, which as explained in the text could actually have built up considerably in the Archean in the absence of an oxidative weathering sink. As for whether less Proterozoic organic carbon burial gives rise to less oxidative weathering in the model, yes it does, but primarily because of the short-term negative feedback described whereby the oxidative weathering flux adjust to match the ‘new’ organic carbon burial flux. The point is that the precise organic carbon content of exposed rocks is irrelevant as long as this supply of organic carbon significantly exceeds the burial of new organic carbon (which we argue that it does).

Lines 162-171. Though it is cited earlier in the manuscript, Claire et al. (2006) should be cited here, as they have examined the effect of falling outgassing rates and changing crustal composition in some quantitative detail.

We have added citation to Claire et al. in the first sentence of this paragraph.

Line 172 (and following section). This section seems to argue that sedimentary $\delta^{13}\text{C}$ should remain stable over most of Earth history because net burial of organic carbon must remain constant to balance both the oxygen and carbon budgets. This implies that the “f-ratio”, which is measurement of net burial (i.e burial minus weathering; this is why is traditionally balanced with only volcanic inputs), does indeed remain constant over Earth history, with the model results providing a mechanism for explaining that stability. It is unclear why a contrast has been drawn with previous discussions of the f-ratio (both here and at line 254), which have observed this constancy but not explained it. Is there a disagreement here over definition?

In essence, yes, the conventional definition of the f-ratio in terms of the proportions of new organic carbon burial to carbonate burial is, we argue, misleading. In fact the thing that is more conserved is the ratio of total=new+detrital organic carbon burial to carbonate burial, as recognised by e.g. Andrey Bekker.

Some discussion of the sulfur cycle is called for. Specifically, it would be useful to understand why, when oxygen levels are forced to another steady state (as in Figure 3), the model compensates for changes in organic carbon burial entirely with changes in organic carbon weathering (allowing fixed $\delta^{13}\text{C}_{\text{carb}}$), when changes in pyrite weathering and burial

should also have their own, possibly complex response to pO₂.

We already discuss why the sulfur cycle cannot provide effective negative feedback on oxygen in the Proterozoic regime – it is because the sedimentary reservoir was almost entirely in the reduced form and the sulfur that is oxidized at the surface is soon reduced again before burial.

Lines 200-204. Both the Holland (2002) and Kump & Barley (2007) argue that these processes would have led to lower rates of reductant input after the GOE, not higher.

Indeed. What we meant here was higher relative to today (not relative to pre-GOE), but it wasn't very clearly written. We have reworded to: "Higher inputs of reductant relative to today – either due to higher mantle fluxes⁵⁰, ongoing crustal oxidation during metamorphism hence more reduced metamorphic fluxes⁴⁰, or greater submarine volcanism resulting in more reduced volatiles⁵¹ – would affect the stability of an oxygen-sink-controlled feedback regime. However, we suggest these extra reductant inputs had largely declined by the Great Oxidation Event or at latest the end of the Lomagundi excursion."

Lines 212-218. See comment on Lines 78-85. A more detailed discussion of elimination of the terrestrial biosphere, and its effects on both NPP and subsequent remineralization and nutrient mobilization processes, would be helpful.

We have now included a range of estimates for the contribution of terrestrial matter to current total organic carbon burial, and written this section in terms of the total organic carbon burial corresponding to particular oxygen levels: "Our proposed Proterozoic oxygen regulation mechanism requires that marine organic carbon burial remained at least ~25% of today's total organic carbon burial flux after the Great Oxidation Event. Marine-derived organic carbon is estimated to comprise half^{11,13} to two-thirds⁵⁴ of today's total organic carbon burial flux (the remainder being derived from land plants). Hence marine organic carbon burial must have remained at least ~40-50% of its present value since the Great Oxidation Event."

Line 224. The model presented here stabilizes at 0.1 PAL, and a clear dynamical explanation is given. However, are the parameterizations robust enough to distinguish between stability at 0.1 and 0.001 PAL, especially given the arbitrary nature of the chosen organic carbon burial rate? The argument provided at lines 237-251 is more grounded in well-constrained observations (i.e., Johnson et al. 2014), though it should be noted that this study did not conclude pO₂ was necessarily as high as 10% after the GOE.

Yes, the parameterizations are robust enough to distinguish between stability at 0.1 and 0.001 PAL. We show results for a full range of organic carbon burial rates below present, including the robust prediction that if organic carbon burial is less than the volcanic source of reduced gases then the Great Oxidation will be reversed – and that this occurs at ~0.01 PAL. Our sensitivity analysis (Fig. 3) does not fundamentally alter this prediction. Even with the most sensitive oxidative weathering parameterisation (red line in Fig. 3a) stability breaks down below ~0.01 PAL.

Lines 252-263. Please see comment for line 172.

This paragraph has been reworked to clarify where we are referring to 'new' rather than detrital or total organic carbon burial.

Lines 274-284. This is an important paragraph, and should be discussed earlier in the manuscript. Maintaining low organic carbon burial is key to their model, but the specified burial rate (50% of modern) has not been clearly explained as function of land plant evolution. The cited model (Lenton et al. 2016) does show that the appearance of land plants can lead to increased oxygen levels in the COPSE model, but their pre-Devonian pO₂ value is of order 20% PAL, substantially higher than in the model here, and many redox proxies should show a systematic difference between the early Paleozoic and the Proterozoic (see Lyons et al. 2014 for a summary). Though a factor of 2 is potentially justifiable, this important factor requires further discussion, and should be articulated clearly earlier in the paper.

These issues are now discussed earlier in the manuscript, and we have now expanded the concluding sentence to acknowledge that there could have been some oxygen rise in the Neoproterozoic: “The rise of oxygen to present levels simply required an increase in organic carbon burial rate, which may have begun in the Neoproterozoic Era¹⁰, but in our model was not completed until the mid-Paleozoic Era with the colonization of the land by plants and fungi, liberating nutrient phosphorus from rocks and producing high C/P material for burial^{11,13}.”

--Thomas Laakso

We thank Thomas for the thorough review, and have added an Acknowledgement.

Alterations to address the Editor’s checklist

We added a sentence to the end of the Introduction to ensure that the last paragraph contains a brief summary of both the results and the conclusions: “Furthermore we show that this mechanism makes the carbonate carbon isotope ($\delta^{13}\text{C}$) record insensitive to changes in organic carbon burial rate, thus explaining the lack of secular trend in the Precambrian $\delta^{13}\text{C}$ record.”

The number of references has been restricted to 70.

A (no) conflict of interests statement has been added.

The figure legends have been grouped before the tables.

The figures have been refined in Adobe Illustrator to match the journal requirements, including e.g. yr⁻¹ (rather than /yr) notation.

REVIEWERS' COMMENTS:

Reviewer #1 (Remarks to the Author):

I believe all my comments were successfully addressed and I am completely satisfied with the revised version of the manuscript. I do not have further comments and look forward to refer to this paper.

Best regards,

Andrey Bekker

Reviewer #3 (Remarks to the Author):

This revised manuscript addresses the major issues raised in my previous review and should be published.

The primary concern in my original review was some overlap between my own model of Proterozoic geochemical cycling, also based on a less-than-modern rate of primary production, balanced by oxygen-sensitive weathering. The authors have noted those results in these edits. They have also included an important reference to Chang & Berner's experimental work on oxidative weathering, which also found a square-root dependence of the weathering rate on pO_2 (albeit under a very narrow range of conditions!). These changes are much appreciated. As the authors point out, the inclusion of tectonic cycling is also a major difference between our two models, and that distinction makes a comparison of their results worthwhile.

Ultimately (as noted in my previous review), this study uses a much-improved parameterization of the key weathering process, and this result alone is worthy of publication.

More generally, this revision has made the proposed geochemical dynamics much easier to follow. In particular, the dynamical differences between the Archean, Proterozoic, and Phanerozoic have been more clearly articulated early in the paper -- in particular, the additions at p.4, l.85, p.6, l.142, and throughout p.10. I recognize that these ideas were in the original manuscript, particularly in Figure 3, but they are easier to follow after the edits.

--Tom Laakso